evolution/computer modelling and simulation

phylogeography, Bayesian phylogenetics, language evolution

**Author for correspondence:**
Nico Neureiter
e-mail: nico.neureiter@uzh.ch

# Can Bayesian phylogeography reconstruct migrations and expansions in linguistic evolution?

Nico Neureiter[1,2], Peter Ranacher[1,2], Rik van Gijn[3,5], Balthasar Bickel[1,3,4] and Robert Weibel[1,2,4]

[1]University Research Priority Program (URPP) Language and Space,
[2]Department of Geography, [3]Department of Comparative Language Science, and [4]Center for the Interdisciplinary Study of Language Evolution (ISLE), University of Zurich, Zurich, Switzerland
[5]Leiden University Centre for Linguistics, Leiden, The Netherlands

NN, 0000-0002-3719-2259; PR, 0000-0002-8680-4063; RvG, 0000-0001-9911-2907; BB, 0000-0002-9087-0565; RW, 0000-0002-2425-0077

Bayesian phylogeography has been used in historical linguistics to reconstruct homelands and expansions of language families, but the reliability of these reconstructions has remained unclear. We contribute to this discussion with a simulation study where we distinguish two types of spatial processes: *migration*, where populations or languages leave one place for another, and *expansion*, where populations or languages gradually expand their territory. We simulate migration and expansion in two scenarios with varying degrees of spatial directional trends and evaluate the performance of state-of-the-art phylogeographic methods. Our results show that these methods fail to reconstruct migrations, but work surprisingly well on expansions, even under severe directional trends. We demonstrate that migrations and expansions have typical phylogenetic and spatial patterns, which in the one case inhibit and in the other facilitate phylogeographic reconstruction. Furthermore, we propose descriptive statistics to identify whether a real sample of languages, their relationship and spatial distribution, better fits a migration or an expansion scenario. Bringing together the results of the simulation study and theoretical arguments, we make recommendations for assessing the adequacy of phylogeographic models to reconstruct the spatial evolution of languages.

## 1. Introduction

Is it possible to reconstruct the place of origin of a language family just by looking at the current geographical distribution of the

extant languages? This certainly seems to be a difficult task. A common way to reconstruct the spread of cultural traits is through diffusion models [1,2], originally applied in ecology [3], which use historical data to map out spatial dispersal over time. But in the absence of historical data, as is usually the case for the analysis of language spread, diffusion models would naively place the origin at the centre of mass of the present-day locations.

As an alternative, it is sometimes proposed that the homeland of a language family lies in the area of maximal diversity, by analogy with patterns in biological evolution [4]. However, the mechanisms of language diversification are subject to additional social and environmental factors that challenge this model [5–9]. Also, there is no mechanistic link between rates of diversification and homeland location. We cannot exclude a slowdown of diversification rates caused by spatial saturation and stable coexistence in the homeland. Nor can we exclude a loss of diversity in the homeland due to later spreads. This makes inferences of homeland locations from diversity patterns unreliable.

Given these shortcomings of previous approaches, Bayesian phylogeography promises a clear improvement for reconstructing evolutionary processes: it builds on reconstructed phylogenetic trees, which reflect how languages are related and when they split. Based on the tree and the present-day locations of the languages, a phylogeographic reconstruction attempts to infer the locations of ancestor languages in the past. Indeed, Bayesian phylogeography has been used to reconstruct the spread of several language families, for example, Arawak [10], Indo-European [11], Bantu [12], Pama-Nyungan [13] or Sino-Tibetan [14]. Some of these reconstructions were corroborated by archaeological evidence and previous results [12,13], while others challenged existing hypotheses [11]. This raises an obvious question: Do these models have the scientific authority to challenge established views on history?

Like all model-based inference, phylogeographic methods are based on assumptions. It is crucial to understand in what cases these assumptions are violated and whether such violations lead to errors in the reconstruction. We see a lack of literature discussing the assumptions and biases of Bayesian phylogeography, in particular regarding what kind of spatio-temporal processes can actually be reconstructed. These issues are particularly important in the case of linguistic phylogeography, where historical data is usually scarce and ambiguous.

In this paper, we evaluate the adequacy of Bayesian phylogeographic methods for reconstructing the spatial evolution of languages. We simulate languages spreading in space and splitting into new languages, based on different historical scenarios. The simulations output recent locations of the languages and a phylogenetic tree, representing their genealogical relationship. The locations and the tree serve as an input to a phylogeographic reconstruction. We test whether this reconstruction can infer the locations of the simulated ancestor languages. In particular, our evaluation metrics are based on the error between the reconstructed and simulated root location (i.e. homeland). Of course, the remainder of the process, what happened between the root and the tips, is of interest as well, but focusing on the root simplifies the quantification of the reconstruction error and is indicative of the quality of the whole reconstruction. Furthermore, reconstructing the homeland of a language family takes a very prominent role in many studies in historical linguistics (e.g. [10,11,15]).

A simulation-based evaluation has two main advantages over an empirical approach: (i) we can perfectly evaluate the reconstructions, since the exact simulated migrations are known. Such knowledge of a ground truth is scarce in historical scenarios and generally when working with empirical data. (ii) We can control all parameters of the process, and this allows us to draw conclusions about the conditions under which the reconstruction works.

We distinguish between two historical processes—migration and expansion—and for each of them we evaluate the reconstruction quality under varying degrees of directional trends. For the purposes of the simulation study, we define migration as a random walk (RW) process, where populations or languages leave a place and move to another place, and expansion as a grid-based region-growing process, where populations or languages occupy cells in a geographical grid and randomly expand to neighbouring cells. The importance of migration and expansion processes in human history has been a source of debate in archaeology and historical linguistics and is sometimes associated with entire migrationist and diffusionist research traditions that emphasize one process over the other [16,17]. In this study, we examine the specific impact of these two processes on phylogeographical reconstruction success.

Historically, we understand *migration* as the permanent movement of entire populations or languages to inhabit a new territory, separate from the one in which they were previously based. Potential causes for large-scale migrations include: environmental push and pull factors, such as climatic changes, migrations of game or the search for more fertile lands. For example, Černý *et al.* [18] discuss Chadic migrations in the context of climatic changes at the end of the African humid period. Migration, as we describe it here, is most often a demic process, i.e. the transfer of languages results from a transfer of populations, but

there are cultural equivalents, where a language is pushed out of its homeland by another one through a process of language shift (and ends up elsewhere). An example is the displacement of the Celtic languages from most of mainland Europe (except Brittany) to the British Isles. What is critical from the point of phylogeographical modelling is that the languages have completely left their homeland.

The term *expansion* describes a series of small-scale movements of people (also called demic diffusion or dispersal) or new people adopting the language (cultural diffusion), accumulating to a slow but steady spread of a language. A common explanation for major historical expansions is population pressure, where growing populations require more resources and territory. Specifically, this explanation has led to the proposal of the *farming/language dispersal hypothesis* [19], positing that 'the present-day distributions of many of the world's […] language families can be traced back to the early developments and dispersals of farming'. In this context, the spreading of many large language families has been described as expansions, including the Bantu, Arawak or Sino-Tibetan languages [19].

We further distinguish between language spread with and without a *directional trend*. While directional trends in migrations typically arise from an intrinsic bias in one direction, expansions also often show directional trends emerging from geographical or ecological constraints, such as oceans, mountains or deserts. When languages expand, but restrictions only allow an expansion in one direction, the resulting spread will follow a directional trend. The reconstruction of migrations or expansions under directional trends seems particularly challenging: how can we know where a directed movement of multiple languages over a long time period started if we only know their current locations? If the whole language family is involved in this movement, the reconstruction seems outright impossible. If only some of the languages were involved, the remaining (sedentary) languages might still uncover the true location of the homeland.

We have seen that both expansion and migration can be driven by demic or cultural processes. Bayesian phylogeographic modelling is blind to the differences between these processes: reconstruction is based on a model of points moving in space. If the reconstruction is based on genetic (demic) data, these points may be readily interpreted as populations. In our case, they represent languages, which disperse through both population movements and through language shifts. Which of these actually was the driver in a specific setting can only be a matter of interpretation in phylogeographic studies. As such, it is not central to this article, where we evaluate the performance of phylogeographic reconstructions on their own.

In addition to the dispersal of languages over time and space, cultural diffusion can also refer to the spread of linguistic features between languages (i.e. the borrowing of words or other properties of a language). Borrowing is a known misspecification of phylogenetic models, which assume that descent with modification along a tree is the default mode of transmission. The assumption has been widely discussed [20] and also evaluated in a simulation study [21]. The current consensus is that the tree model is a well-justified default and that confounds from borrowing can be controlled reasonably well, although of course challenges remain. In this article, we focus on the geographical component of phylogeographic models and assume a setting where the phylogenetic tree is faithfully reconstructed. This allows us to single out errors that are specifically due to misspecification in the geographical model rather than a lack of control for borrowing. Empiricists can aim for a similar setting in cases where a reconstructed phylogenetic tree of the languages under investigation is available (a range of such trees is available on D-PLACE [22]). Otherwise, the phylogeny needs to be inferred from linguistic data (typically cognate sets). Either way, these reconstructions are not guaranteed to reflect the true genealogy of languages and need to be subject to scrutiny as well.

We implement the concepts of migration and expansion in two corresponding simulation scenarios, both with varying degrees of directional trends. We implement the migration simulation (MigSim) in the form of directional RWs. In a RW process, particles (in this case representing languages or populations) move in space stochastically. In the case of directional RWs, this stochastic movement has an intrinsic bias in one direction. This model has been used in movement analysis to study animal migrations [23,24] or the spread of viruses [25].

We then further implement a simulation of historical linguistic expansions, with and without directional trend. In this scenario, languages are represented by geographical areas, rather than point locations. These areas expand and once they reach a certain size, the language splits up. While in reality there are languages with huge and sometimes scarcely populated ranges, the size constraint generally seems to be a reasonable assumption, since large geographical areas make contact between some of the people in the population less likely and at some point, this leads to diversification [26]. After the split, the two resulting languages continue to grow and split separately. Geographical constraints and clashes with other populations can force the expansion to proceed into a predominant direction, resulting in a directional trend.

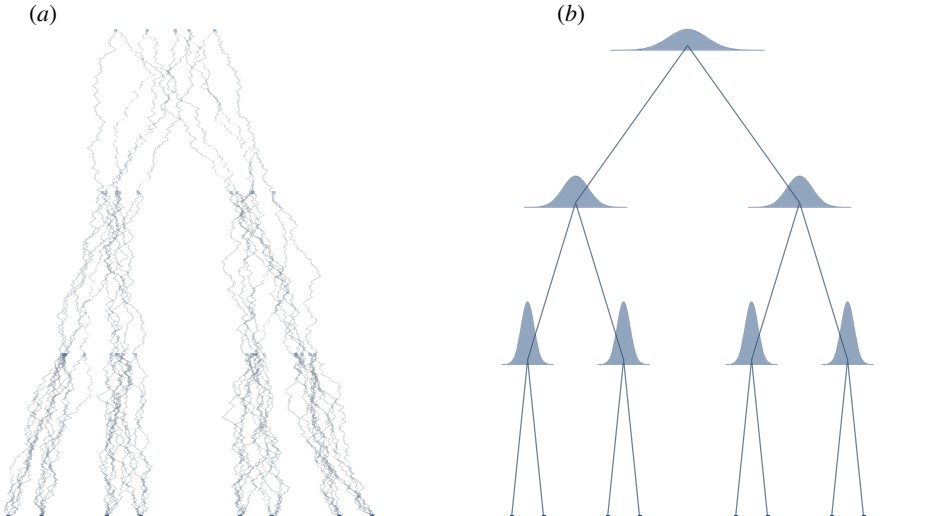

*(a)*          *(b)*

**Figure 1.** For a given phylogenetic tree different random walks can explain the data. (*a*) Five examples of random walks (in one spatial dimension), leading to the same tip locations. (*b*) Marginal posterior distributions of each internal node location under this random walk model.

We will show how these two simulations of migration and expansion processes lead to quite distinct spatial patterns, resulting in strong differences in the accuracy of the phylogeographic reconstruction.

## 2. Phylogeographic modelling

Relationships between languages have been modelled as trees since the nineteenth century, but the field is increasingly adopting phylogenetic methods from evolutionary biology to model language evolution [27–29]. These methods reveal both the pattern of how organisms split, i.e. the topology of the phylogenetic tree, and the rate at which organisms split, i.e. the branch lengths in the tree. In addition, they allow the inference of trait evolution—the change of properties along the tree [30].

State-of-the-art methods of phylogenetic analysis rely on computational algorithms and statistical models, in particular, Bayesian inference and Markov chain Monte Carlo sampling (MCMC) [31]. Bayesian phylogenetic analysis proposes possible evolutionary histories, and evaluates these against the likelihood of the data and prior beliefs. The analysis yields a posterior distribution and, thus, captures the inherent uncertainty of the evolutionary process [31]. There are several computer programs for Bayesian phylogenetic analysis, the most frequently used being BEAST [32] and MrBayes [33]. In a recent study, Wichmann & Rama [34] compared different phylogeographic software packages and found 'no radical differences' in their performance.

Bayesian phylogeography [35,36] aims to infer the spatial expansion of a phylogenetic tree, that is, the spatial location of all ancestral nodes (including the root) and the rate at which organisms or languages change their location—the diffusion rate. The spatial locations make up the path of the expansion. The diffusion rate scales the expansion to the geographical and temporal context. Originally, Bayesian phylogeographic analysis was developed to reconstruct the spatial diffusion of viruses [35]. Recently, it has also been applied extensively in linguistics to explore the historic expansion of language families [10–14].

Phylogeography can be viewed as a special case of trait evolution, where the changing property is the spatial location, modelled as either a discrete site [36] or a continuous position on a plane [10,35] or sphere [37]. While discrete phylogeographic models are important, e.g. in epidemiology [38,39], they have not been used thus far in historical linguistic contexts, which is why we focus on continuous models: we briefly outline how these models allow inferences and summarize their main properties.

Bayesian phylogeographic models accommodate a Brownian diffusion (BD) [40] process along the phylogenetic tree. Starting at a potential spatial location of the root, the BD moves down the branches of the tree, proposing locations for all ancestral nodes in turn, until reaching the observed tips [25]. We show samples from such a BD process along a fixed tree, ending in fixed tip locations in figure 1*a* and the resulting distributions at the internal nodes in figure 1*b*. We can see that the uncertainty is 0 at the tip locations and increases as we move further back in time. In two-dimensional space, the likelihood of moving down a branch is modelled as a bivariate normal distribution with a

displacement mean and variance. The mean is usually set to zero, i.e. there is no directional trend in the expansion. The variance is proportional to the rate of diffusion times the branch length, which is the amount of time needed to reach the next ancestral node. In an MCMC, possible spatial expansions are proposed and evaluated against their likelihood. Intuitively, a spatial expansion yields a high likelihood if it minimizes the distance from the root through all ancestral nodes to the observed tips. In addition to that, fossils—known spatial locations of extinct organisms (or languages, in our case) [41]—can help to place ancestral nodes, and thus, improve reconstruction.

There are three state-of-the-art phylogeographic models, all of which build on the concept of BD as outlined above:

— RW models assume the diffusion rate to be constant along the entire phylogeny. This implies that the expansion in space is homogeneous [35].
— Relaxed random walk (RRW) models allow the diffusion rate to vary and the dynamics of the expansion to change along the tree. In RRW, bursts of rapid migrations can be followed by phases where a population remains stationary [35]. Since a dynamic expansion is realistic in many application examples [12,35,42], RRW models are the de facto standard for phylogeographic reconstruction.
— Directional random walk models allow the displacement mean to differ from zero. This introduces a directional trend (or directional bias), such that the expansion is free to prefer a particular spatial direction. The constant directional random walk (CDRW) [43] assumes the displacement mean to be constant over the entire expansion, whereas in the relaxed directional random walk (RDRW) it is allowed to vary [25]. In theory, directional random walk models should reconstruct directed migrations, but they rely on fossils to inform the inference [25].

In this article, we evaluate the performance of phylogeographic models to correctly infer spatial patterns in language phylogenies resulting from either migrations or expansions, with particular attention to patterns with a directional trend. RW and RRW do not model a directional trend, suggesting that inference will favour a radial expansion from a location somewhere in the centre of the observed distribution of the tips, even in the presence of a trend. While CDRW and RDRW models should be able to infer directional bias, it is unclear to what extent they allow the reconstruction of migrations or expansions without historical information in the form of fossils.

We evaluate the performance of phylogeographic reconstructions in a simulation study. In phylogenetics, simulation studies are widely used to validate the consistency of new models under model assumptions (e.g. [25,35,44]). In contrast to this, we simulate historically motivated scenarios of migrations and expansions, which intentionally violate some of the model assumptions. In that respect, our study is similar to [21], which explores the effect of horizontal transmission on phylogenetic reconstruction, a process not accounted for in basic phylogenetic inference.

# 3. Methods

We propose a simulation-based procedure to evaluate the performance of Bayesian phylogeographic methods, consisting of three steps:

(i) *Simulate* a phylogenetic tree and movement of the languages in space.
(ii) *Reconstruct* the movement based on the simulated tree and tip locations using phylogeographic analysis.
(iii) *Evaluate* the results by comparing the reconstructed movements with the originally simulated ones.

In the first step, we randomly generate phylogenetic trees and the spread of the corresponding languages in space and time. By changing the way in which we simulate the spread, we can test the sensitivity of the reconstruction to different movement scenarios. In particular, we are interested in the performance of phylogeographic methods in movement scenarios with directional trends. In this section, we explain the exact methodology behind each of these three steps.

## 3.1. Simulations

We simulate the evolution of related languages in space as two temporal processes: movement and diversification. That is, the simulated languages move in space and randomly split to form descendant languages, which then continue to move and diversify themselves. We focus on the movement

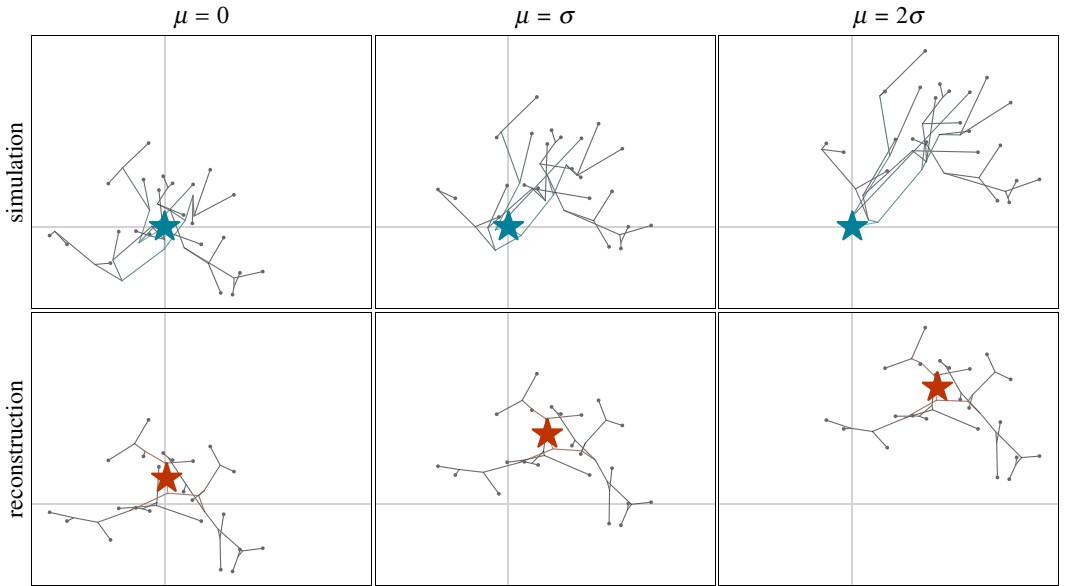

**Figure 2.** Examples of three simulations and corresponding reconstructions. The top row shows the simulated trees plotted in space with the root marked by a blue star. In the bottom row, the reconstructed tree can be seen with the root in red. The columns represent different levels of directional trend in the simulation. The trend increases from left to right ($\mu = 0$, $\sigma$, $2\sigma$).

process, hence we do not simulate linguistic features and only simulate diversification of languages in the form of a phylogenetic tree, representing their taxonomic relationship.

We introduce two simulation scenarios corresponding to the two historical processes of migration and expansion.

### 3.1.1. Migration simulations

We model migrations as directional random walks, where languages are represented by points in space. These points move stochastically with a bias in one direction. The direction and strength of this bias is controlled by the parameter $\mu$. Examples for different levels of $\mu$ are displayed in figure 2. Other parameters of the MigSim are $T$, the total time span of the simulation (from the first split to the current time); $N$ the expected number of leaves (i.e. sampled languages) in the tree; and $\sigma$, the expected distance covered due to undirected movements over the whole expansion period. Here, we set these parameters to values that seem realistic for the expansion of a language family ($T = 5000$ years, $N = 100$ nodes and $\sigma = 2000$ km). However, we want to emphasize that the exact values do not matter for our findings. The results show more generally how the reconstruction quality changes when increasing $\mu$, for a fixed $\sigma$. A sensitivity analysis on the number of nodes shows that varying the tree size does not change our findings (see electronic supplementary material, S6).

We implement directional random walks in discrete time steps of duration $\Delta_t$ (in our experiments set to 1 year). In every time step, each language makes a random move according to a Gaussian distribution

$$X_{t+\Delta_t} \sim \mathcal{N}(X_t + \mu_{\text{step}}, \sigma_{\text{step}}^2). \tag{3.1}$$

The free parameters in this process are the step mean $\mu_{\text{step}}$ and the step variance $\sigma_{\text{step}}^2$. Since these steps are arbitrary units of our simulation, we aggregate them into more meaningful quantities: the total bias $\mu$ and total standard deviation (or total expected diffusion distance) $\sigma$, defined as

$$\mu = T \cdot \mu_{\text{step}} \tag{3.2}$$

and

$$\sigma = \sqrt{T} \cdot \sigma_{\text{step}}. \tag{3.3}$$

This gives rise to a reformulated step distribution

$$X_{t+\Delta_t} \sim \mathcal{N}\left(X_t + \mu/T, \sigma^2/T\right). \tag{3.4}$$

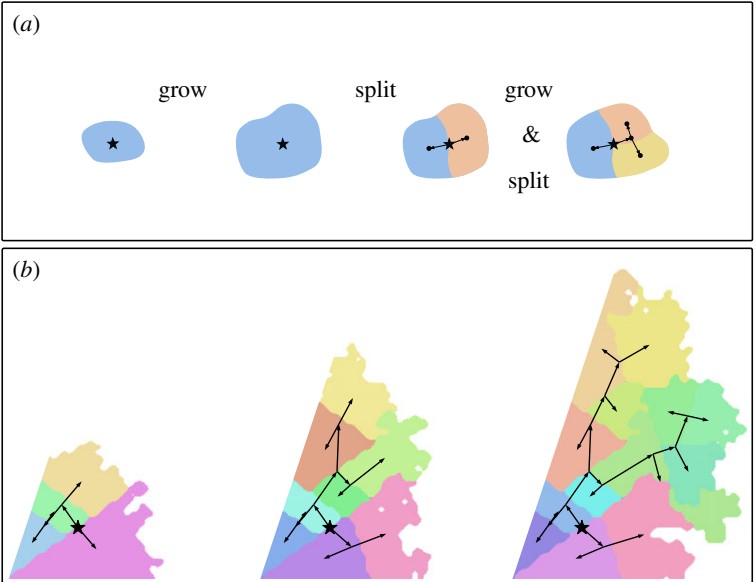

**Figure 3.** Two illustrations of the expansion simulation: (*a*) every language is represented by a coloured area, which over time grows into free surrounding areas and splits up into new languages. The result is a phylogenetic and geographical expansion, as we can further see in (*b*): the expansion started at the black star and grows into new free space (visualized at three time points). Free space in this case is only available in a 72° sector. The phylogenetic tree is visualized by the black edges, leading from the root (black star) to the different areas, representing the extant languages.

At the same time, each language has a certain probability to split into two new languages, which then continue to undergo independent random walks. In order to simulate historical data from extinct languages (fossils), each language has a certain probability to go extinct. This is a common birth–death process, which is controlled by the birth rate $\lambda$ and death rate $\nu$. We set $\nu = \ln N / 4T = 0.00023$ and $\lambda = 5\nu = 0.00115$, which after $T = 5000$ years results in an expected number of languages of $N = 100$. Including a death rate in the model allows us to use extinct languages in some of the experiment settings (see electronic supplementary material, S1). To ensure comparability between the scenarios, we remove outliers from the simulation results. We consider a result as an outlier if the number of extant languages is below 40 or above 200.

### 3.1.2. Expansion simulations

We propose expansion simulations (ExpSim), which can also be subject to directional trends; however, not through an inherent directional bias, but through geographical constraints, forcing an expansion in one direction. Geographical constraints could be barriers, such as mountains or oceans, which are difficult to traverse, or simply disfavoured regions, where for example an important crop does not grow or land is ascribed negative cultural values. We capture this constrained expansion scenario in a simulation based on the following ideas (figure 3): languages are represented by areas consisting of cells of a grid. These areas randomly expand over time into free neighbouring cells, i.e. cells that are not occupied by another language or blocked by geographical constraints. In a sensitivity analysis (see electronic supplementary material, S6), we relax this assumption and allow up to three languages to occupy the same cell. With increasing area a language becomes more likely to split into two new languages, which in turn continue to expand separately. This simulation carries some resemblance to the model presented by Gavin *et al.* [26], but with crucial differences: while their model simulates language areas sequentially, the ExpSim model simulates splits, inducing a tree structure in the simulated languages.

As illustrated in figure 3, the ExpSim simulation is defined by two parallel processes: growing and splitting of areas. Growing is controlled by a parameter $p_{grow}$. At every step of the simulation, an area adds a random cell from its neighbourhood with probability $p_{grow}$. We implement heterogeneity in the growth rate by drawing $p_{grow}$ uniformly from the interval [0, 1] for each language. At a certain randomly chosen size a language splits up into two. In our experiments, we choose the split-size uniformly at random and for every language independently between 70 and 100 cells. Again, the exact values do not significantly influence the results. We chose the distribution of the split-size to

roughly match the expected tree size in the MigSim simulations. Again, we run the simulation for 5000 steps. In order to make space and time comparable, we define a step to correspond to 1 year and a cell to correspond to $100 \times 100$ km.

To introduce a directional trend into the simulation, we model geographical constraints. In reality, these constraints (e.g. deserts, mountains or seas) may have arbitrary shapes, but to systematically investigate the properties of constrained expansions we confine the simulations to geometric examples: the first language starts to expand in the corner of a circular sector of angle $\alpha$ and is only allowed to grow within this sector. Leaving the sector completely open ($\alpha = 2\pi$) allows a concentric expansion in all directions. Choosing the sector very tightly ($\alpha \to 0$), forces the expansion to move in one direction out along this narrow sector. In this way, we can control $\alpha$ to achieve different levels of directional trend in the expansion. To compare this trend to the inherent bias parameter in the MigSim scenario, we will introduce a measure for the observed trend in §3.3.

## 3.2. Reconstruction

To evaluate the performance of phylogeography, we start from a setting which is optimistic compared to real-life applications: we fix the phylogenetic tree to the true (simulated) one and perform phylogeographic analysis based on this tree and the tip locations. In a real case study, the tree would have to be reconstructed as well. Fixing the tree is a valid simplification of the experiment, because it allows us to focus specifically on the geographical reconstruction on its own. The result of the analysis is a reconstruction of the locations at the internal nodes of the tree, in particular the root location (the homeland). We perform this phylogeographic analysis in BEAST 1.10.4 [45], where the original RRW model [35] and extensions to directional random walks [25] are implemented.

For the phylogeographic analysis, we test the RRW and the two directional random walk models CDRW and RDRW. All models were set up with variable rates according to a lognormal clock. The prior settings of the analysis are in line with commonly used values in previous analyses (e.g. [11]). Moderate changes in the prior on the diffusion rate and the variance of the lognormal clock did not show any notable effects. In CDRW and RDRW, the prior on the drift parameter (directional bias) heavily influences the reconstruction in cases where we do not provide enough fossils to calibrate the parameter. The effect of the prior ranges from a fall-back to the RRW model, for priors that are very narrow around 0, to a highly variant reconstruction that would allow for directed migrations from any point on the map, for very wide priors.

## 3.3. Evaluation

A Bayesian phylogeographic reconstruction, as the one described above, provides a posterior distribution over possible root locations. We are interested in two properties of this reconstruction: (i) How far do we expect the reconstructed location to be from the true homeland? (ii) How certain is the model about its reconstruction? We measure these two properties using the following metrics:

(i) We measure the Euclidean distance between every location in the posterior distribution and the true root to see how far the reconstruction is off. Doing this per posterior sample and taking the root of the mean squared error gives the RMSE metric

$$\text{RMSE} = \sqrt{\sum_{m=1}^{M} \frac{\left\| \hat{X}_m - X \right\|^2}{M}}. \tag{3.5}$$

Here, $X$ is the true location of the homeland and $\hat{X}_m$ are the reconstructed locations, taken from all posterior samples over multiple simulation runs. The RMSE is composed of systematic errors (bias) and errors due to variance (in the simulation and in the reconstruction). Since we are mainly interested in systematic errors arising from directed migrations we also present the bias of our reconstructions

$$\text{bias} = \left\| \sum_{m=1}^{M} \frac{\hat{X}_m}{M} - X \right\|. \tag{3.6}$$

As an illustration of the behaviour of these two metrics consider the examples shown in figure 2. With growing directional trend $\mu$, the reconstructed root (red star) will be located further away from the

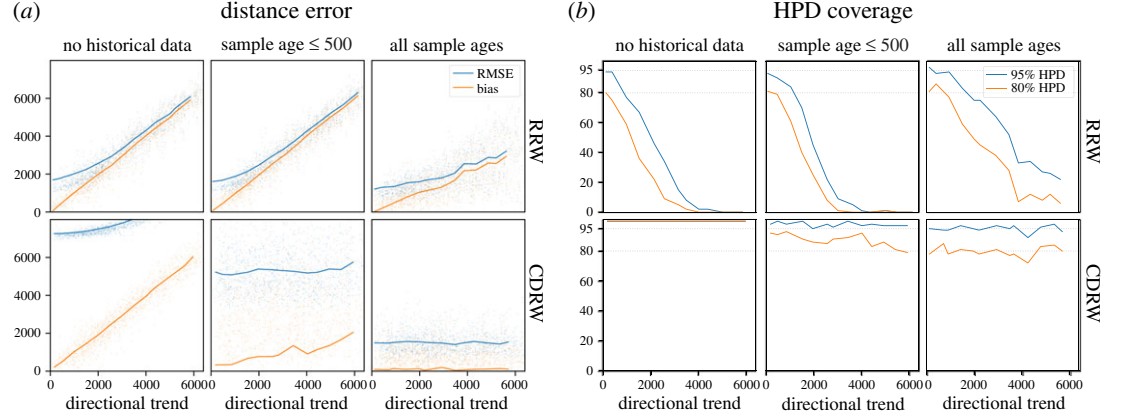

**Figure 4.** The performance of phylogeographic reconstructions of the root based on the MigSim simulations with varying levels of observed trend. (*a*) The RMSE (blue) and the bias (orange) of the reconstruction. The dots represent single simulation runs, the lines interpolate between the average results for a specific setting for $\mu$ (trend). (*b*) The empirical coverage of the 80% (blue) and 95% (orange) highest posterior density (HPD) regions.

simulated one (blue star), causing both the RMSE and the bias to grow. The difference between the two scores would mostly show in scenarios with low directional trend ($\mu$ close to 0), where the bias would approach 0, while the RMSE still shows errors due to variance.

(ii) We measure the Bayesian highest posterior density (HPD) region coverage of the root location for different HPD thresholds (80% and 95%). This entirely ignores the amplitude of the error, but shows whether the uncertainty expressed by the posterior reflects the observed error. If so, the HPD coverage will match the corresponding HPD threshold, i.e. the 95% HPD region should cover the root in 95% of the simulations.

In order to compare these evaluation metrics between the different scenarios, we want to measure directional bias in a unified way. In the case of migrations, the directional bias is introduced through an inherent trend parameter $\mu$, in the expansion simulations it is controlled by the sector angle $\alpha$ (and in a real case study, we usually do not know the factors driving the directed migrations at all). To compare these scenarios, we measure the *observed trend* $\hat{\mu}$, which we defined as the distance between the homeland of an expansion and the mean of the final locations (at presence)

$$\hat{\mu} := \left\| \sum_{n=1}^{N} \frac{x_n}{N} - X \right\|. \tag{3.7}$$

Here, $X$ represents the root of the expansion and $x_n$ are the tip locations (contemporary languages). We use this statistic to make the varying levels of directional trend in the two simulation scenarios comparable: in the MigSim scenario, we simulated a total bias $\mu$ ranging from 0 to 6000 km (in steps of 500 km) and in the ExpSim scenario the sector angle $\alpha$ ranged from $2\pi$ down to $0.2\pi$ (in steps of $0.2\pi$). In both scenarios, these settings resulted in an observed trend $\hat{\mu}$ between 0 and 6000 km (figures 4 and 5).

## 4. Results

We ran experiments based on the migration (MigSim) and expansion (ExpSim) scenarios. In both scenarios, we look at cases with different levels of directional trends and we reconstruct the root using both the standard RRW and the CDRW models. The results for the MigSim and the ExpSim scenarios are presented in figures 4 and 5, respectively. Figures 4*a* and 5*a* show the bias and root mean square error (RMSE) of the reconstruction, indicating how far we expect the reconstructed homeland to be located from the real one. Figures 4*b* and 5*b* depict the 80% and 95% HPD coverage, indicating the overconfidence (coverage is below 80%/95%) or underconfidence (coverage is above 80%/95%) of the model. In what follows we summarize the main results.

In the absence of historical data, directed migrations lead to significant errors in the phylogeographic reconstruction. This can be seen in figure 4*a* in the leftmost column (no historical data): the reconstruction bias increases linearly with the simulated directional trend. If we observe a directional trend of 5000 km, the

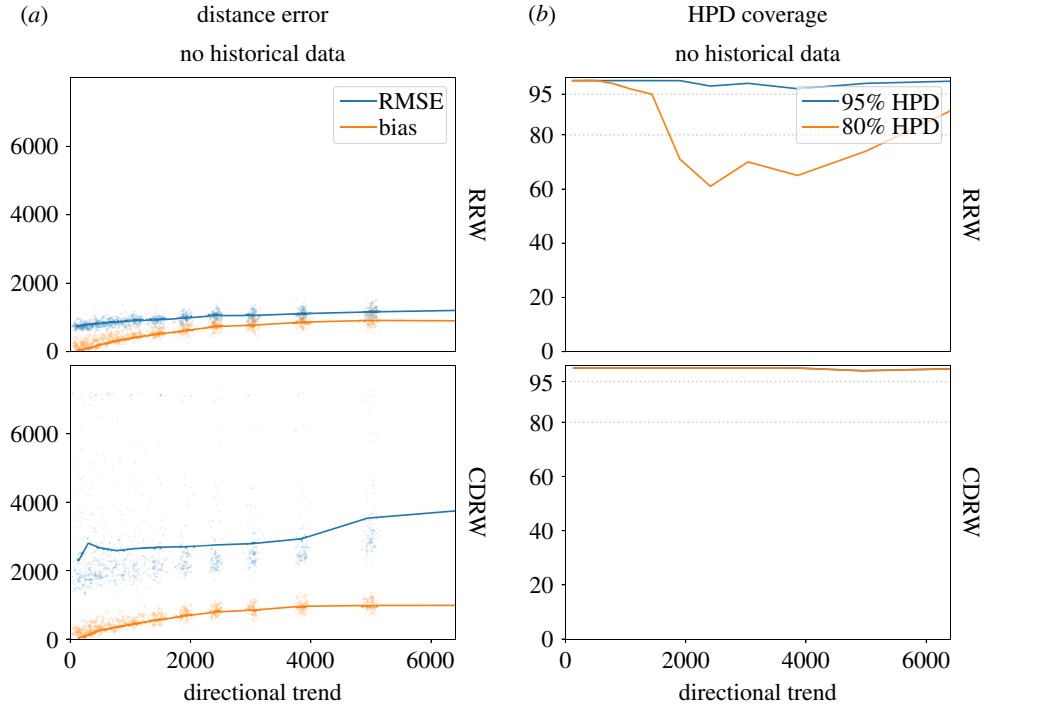

**Figure 5.** The performance of phylogeographic reconstructions of the root based on the ExpSim simulations with varying levels of observed trend. (*a*) The RMSE (blue) and the bias (orange) of the reconstruction. The dots represent single simulation runs, the lines are averages across all runs with a specific setting for $\alpha$ (sector angle). (*b*) The empirical coverage of the 80% (blue) and 95% (orange) highest posterior density (HPD) regions.

reconstruction is off by about 5000 km too. This holds for reconstructions using the RRW (top row) as well as the CDRW (bottom row) model. This is expected, since the CDRW model needs historical data to be calibrated [25]. The RMSE (blue line) of the CDRW reconstruction is even significantly higher, due to a higher variance in the posterior. A look at the HPD coverage statistic shows that this increased variance is an effect of a very agnostic or even underconfident model, where the 95% and even the 80% HPD regions always cover the simulated root. In the RRW model, by contrast, the corresponding HPD coverage approaches 0 with increasing directional trends, as expected in the case of an increasingly significant mismatch between the model assumptions and the data.

Including historical data in the analysis reduces the error in both the RRW and the CDRW model. If we include sampled historical locations from the whole time period, the CDRW model is able to estimate the directional trend from the data, which leads to a bias close to 0 and 80%/95% HPD coverage around the ideally expected values of 80%/95%, respectively (bottom row, right). The reconstruction of the RRW model, on the other hand, only flattens the slope at which the error increases. At an observed trend of 5000 km, the reconstruction bias is still above 2500 km. Furthermore, the model is still overconfident, visible in the drop of the HPD coverage. Finally, we find that including historical locations at shallow time depths of up to 500 years does not significantly improve the reconstruction with either model.

In expansion scenarios (ExpSim), we see a very different picture. Even in the absence of any historical information, the reconstruction error levels off far below what we would expect in the MigSim scenario, as can be seen in the top panel of figure 5*a*. After an initial increase of the reconstruction error, even at an observed directional trend of 7000 km the reconstruction bias does not exceed 1000 km. Since the exact values of these errors (bias and RMSE) are influenced by the scaling parameter (kilometres per cell) and the shape of the simulated constraints, it is the more general results that we want to emphasize: (i) The bias does not increase linearly with the observed trend, but clearly stagnates, and (ii) the estimated HPD regions cover the true homeland very consistently (figure 5*b*). These results turn out to be independent of the phylogeographic model used. Resorting to an explicit directional model (CDRW or RDRW) only increases the RMSE, through a higher variance in the posterior distribution. The results for the RDRW model can be found in electronic supplementary material, S5. We did not simulate fossils in the ExpSim scenario. Even in the absence of historical information, the estimated reconstructions consistently recover the simulated homeland, therefore including fossils was not necessary.

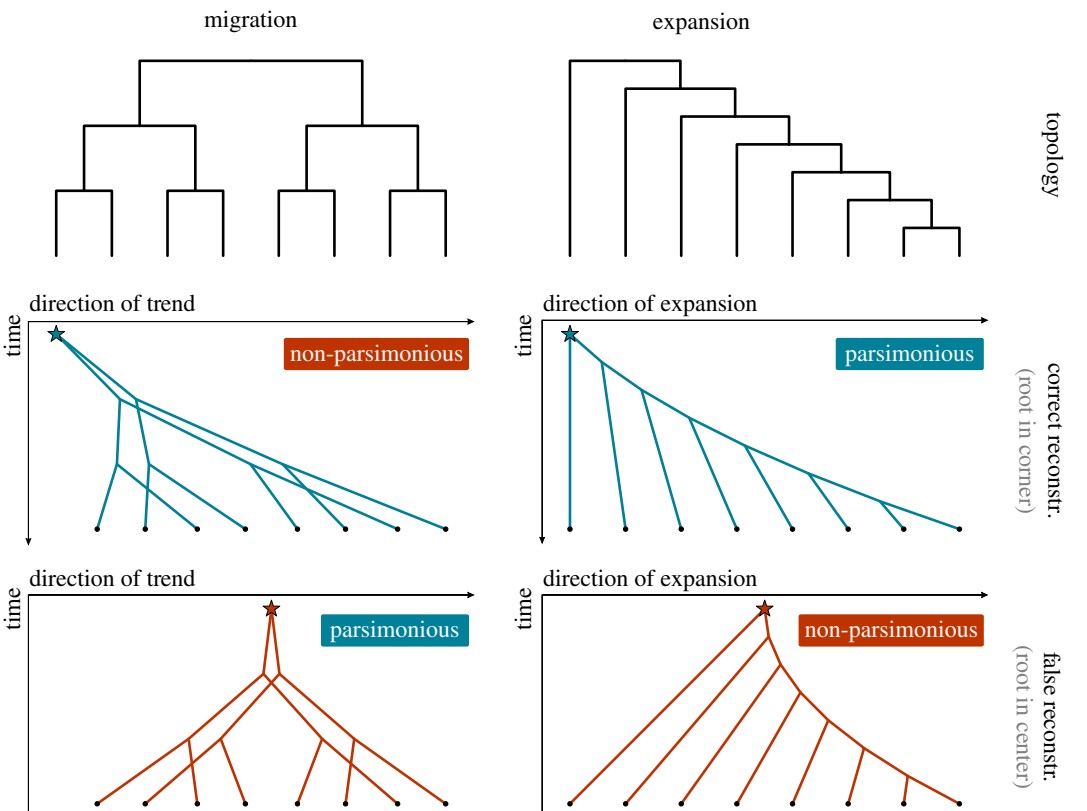

**Figure 6.** Visualizations of two scenarios of directional trend: migration (MigSim) and expansion (ExpSim). In all plots, the *y*-axis represents time. The top row depicts typical tree topologies (balanced versus imbalanced) assuming no further variation in birth and death rates. The middle row shows typical migration patterns in one spatial dimension (parallel migrations versus nested splitting and expansion). The bottom row shows how a hypothetical reconstruction of the homeland at the centre of gravity would lead to more versus less parsimonious reconstructions. In the expansion scenario, the more parsimonious case coincides with the correct reconstruction.

Furthermore, a sensitivity analysis (electronic supplementary material, S6) shows that the tree size does not notably affect the reconstruction quality (electronic supplementary material, S6.1) and that our general findings for the ExpSim scenario still hold if areas are allowed to overlap (electronic supplementary material, S6.2).

# 5. Discussion

In summary, the results of the simulation study reveal that the reconstruction quality is vastly different in different movement scenarios:

— The migration scenarios (MigSim) lead to a severe bias in the reconstruction that grows proportionally with the directional trend in the migration. This bias remains when including recent historical samples and is independent of the tree sizes (see electronic supplementary material, S6.1).
— In expansion scenarios (ExpSim), the model shows some uncertainty about the precise location of the origin, but no severe reconstruction errors, even under strong directional trends. The successful reconstruction does not rely on historical samples, and is robust to overlapping language areas and changes in the tree size (see electronic supplementary material, S6.1 and S6.2).

In what follows, we discuss why phylogeographic methods perform so much better in the ExpSim scenario (§5.1) and we make recommendations for future work (§5.2).

## 5.1. Differences in reconstructability

Why are we able to reconstruct the origin in the expansion scenario, while the same seems to be impossible for the migration scenario? We visualize the differences in figure 6 and show statistics

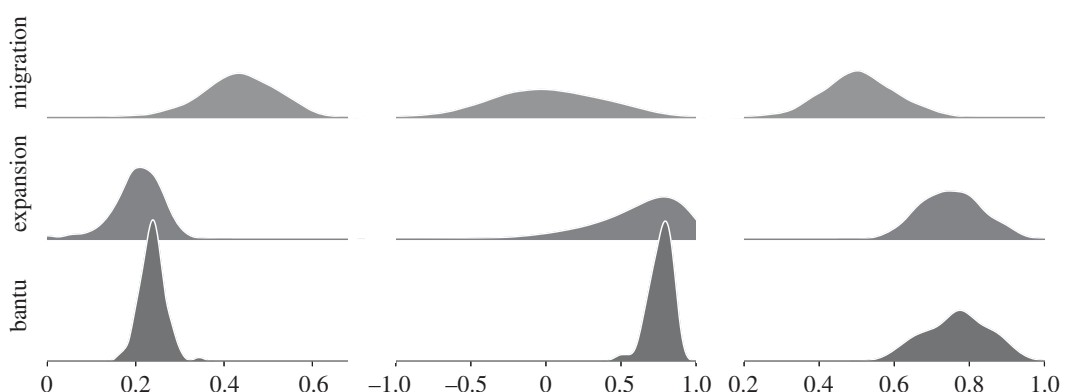

**Figure 7.** Three descriptive statistics (1) clade overlap, (2) diversity-space dependence and (3) tree imbalance (see electronic supplementary material, S7) evaluated on samples from the MigSim simulation, the ExpSim simulation and the posterior distribution of the Bantu phylogeny.

characterizing the different scenarios in figure 7. In a typical migration scenario (figure 6, left column), the diversification and the movement processes are independent. If there are no other factors causing variation in the birth or death rates, we expect a balanced tree topology. The migrations follow a random walk along this tree with a bias in one direction, as visualized in the middle row of figure 6. Without any historical information, there is no way for a reconstruction algorithm to identify this direction. The most sensible (parsimonious) reconstruction is achieved by a homeland at the centre of mass of the recent locations (figure 6, bottom left).

Expansions lead to different spatial and phylogenetic patterns compared to migrations, even if they show the same level of observed directional trend (right column of figure 6). First, expansions do not cause all languages or language-carrying populations to move in one direction. Actually, every language or population tends to stay stationary and it is only the diversification and growth into new areas that is forced to proceed in one direction due to geographical constraints. In turn, clades which stay stationary are more limited in their space for further expansion and diversification. As a result, the region of the homeland will mostly be populated by clades that stopped to diversify and to migrate a long time ago (compared to the languages and populations at the frontier of the expansion). In figure 6 (right column), we illustrate what such an expansion typically looks like, with an imbalanced tree topology, where languages split off one by one. The languages splitting off earlier remain closer to the homeland, while the expansion continues to spread, causing the directional trend (figure 6, middle right). Figure 6 (bottom right) shows that a reconstruction of the true directed expansion is actually more parsimonious than a spread from a hypothetical central homeland, which would imply multiple parallel migrations of early-split languages towards the left.

We propose three descriptive statistics to characterize the different patterns seen in the MigSim and ExpSim scenarios (see electronic supplementary material, S7 for details): The *clade overlap* score measures how much languages from different clades overlap in space. Under the ExpSim scenario, clades are tightly connected, since the areas are generally stationary and cannot cross each other (this constraint is relaxed in the electronic supplementary material, S6.2; overlap score of $0.20 \pm 0.06$). In the MigSim scenario, languages can move and cross each other freely, leading to an increased clade overlap ($0.43 \pm 0.09$).

*Diversity-space dependence* measures the correlation between the spatial expansion and diversification of a clade. A strong correlation implies that clades expanding faster also diversify faster, resulting in a higher number of languages. This is intended to capture the fact that diversification in the ExpSim scenario is directly governed by the area of a language, while the two processes of spread and diversification are independent in the MigSim model. This leads to a dependence score of $0.60 \pm 0.34$ for ExpSim and $0.02 \pm 0.36$ for MigSim.

Finally, *tree imbalance* quantifies the asymmetry of a tree, i.e. the extent to which lineages in the tree diversify unevenly. Trees generated from the ExpSim scenario are generally imbalanced ($0.75 \pm 0.09$), because the growth of earlier clades is inhibited through geographical constraints and other clades (as described above and visualized in figure 6). In the MigSim scenario, the tree shape follows a common birth–death process, leading to a relatively balanced tree topology ($0.50 \pm 0.10$).

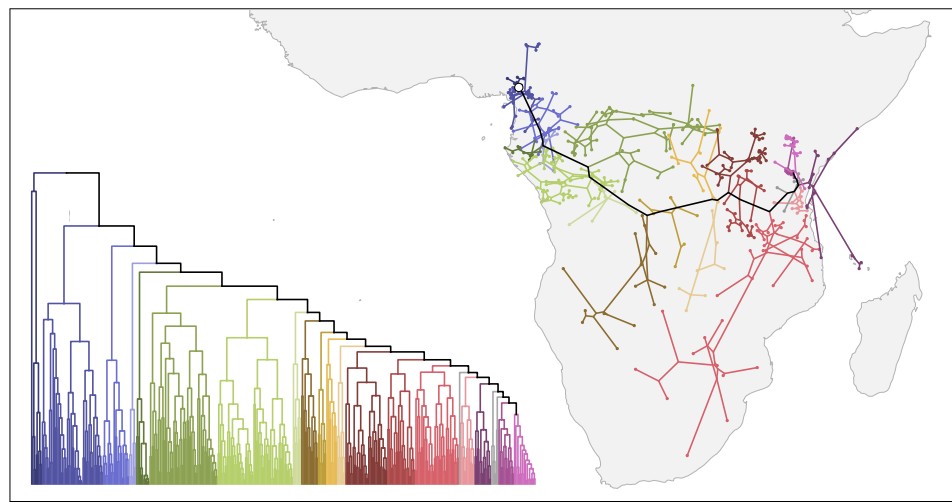

**Figure 8.** The phylogeny and spatial spread of the Bantu languages according to [12]. The colours mark the clades, splitting off one after another from the backbone of the expansion.

These descriptive statistics may help to identify whether the dispersal of a language family mainly reflects a migration or an expansion scenario, which in turn indicates whether we can trust the phylogeographic reconstruction. To avoid an influence of the phylogeographic model on the descriptive statistics, and thereby a potentially circular justification, the phylogenetic tree used in the computation of these statistics should be inferred from linguistic data alone (without a geographical model).

Beyond this, we also want to highlight a basic property of phylogeographic reconstructions (using the RRW model): The reconstructed root will always fall within the convex hull of the sampled locations. As shown in [37], the mean location of the internal nodes can be expressed as a weighted average of the sampled locations, directly implying that they must be within the bounds of the convex hull. Clearly, phylogeography is not an adequate reconstruction tool if a migration from outside the sampled area (or a loss of the languages in the area of the homeland) is a plausible hypothesis.

## 5.2. Implications for linguistic phylogeography

In the remainder of this section, we explore the application possibilities and restrictions of current phylogeographic methods, and make recommendations for judging the adequacy of phylogeographic models to reconstruct the spatial evolution of languages. The simulation study presented in this paper shows that phylogeography provides faithful reconstructions of expansions, but fails severely on directed migrations. To translate these findings into practical decisions for the reconstruction of a real language family, researchers need to judge whether the history of those languages suggests an expansion or a migration scenario. If a plausible hypothesis about the history of the languages involves directed migrations, phylogeography is not an appropriate tool for reconstruction or hypothesis testing.

We now discuss the Bantu languages as an example of a linguistic expansion with an existing phylogeographic analysis (Grollemund *et al.* [12], available as part of D-PLACE [22]). The spread of the Bantu languages goes hand in hand with one of the major expansions of agriculture on the planet. As the farming/language dispersal hypothesis posits [19], the availability of agriculture explains a continuous spread of the Bantu people and languages into most of southern Africa. The geographical constraints of this dispersal are roughly defined by oceans, deserts (Sahara, Kalahari), and land occupied by related Niger-Congo societies (West Africa). In fact, the Bantu languages nowadays extend over most of the geographical area bounded by these constraints. The phylogeny of the Bantu languages exhibits similar patterns to those of the expansion (ExpSim), and not those of a migration scenario (MigSim; figure 7): The phylogenetic tree is imbalanced (imbalance score of $0.77 \pm 0.08$), with little clade overlap (overlap score of $0.24 \pm 0.03$) and the clades that split off and became stationary also diversified less than the rest of the tree, which was able to expand into a wider territory (leading to a diversity-space dependence score of $0.76 \pm 0.08$). This process is quite clearly visible in the reconstructed expansion (figure 8). All these indications would support the notion that the spread of the Bantu languages is well described as an expansion scenario and, indeed, the phylogeographic reconstruction in [12] very accurately matches the previously proposed homeland in the Grassfields

region in Cameroon (supported by strong scientific consensus, based on archaeological [46,47], linguistic [48] and genetic evidence [49]). Similar patterns where evidence is suggestive of multiple nested splits corresponding to chained movements into new territories are found all over the globe. For example, the Austronesian expansion is also characterized by an imbalanced tree topology and multiple successive waves of expansions towards the east [50]. In the Austronesian case, geographical constraints, coupled with the development of seafaring technology seem to have determined the pace and direction of the expansion [6].

If the history of the languages under study does not align with the expansion scenario, we cannot warrant the same optimism expressed above. If the history involved directed migrations we would even expect the reconstruction to fail. This includes cases of demic migrations as well as scenarios where languages were replaced and pushed out of their initial territory in a language shift. The latter is clearly observable in the case of the Celtic languages, which once extended across vast areas of Europe. But since the population in these areas shifted to Germanic and Romance languages, Celtic is now only spoken in northwestern France (Brittany) and on the British Isles, making a phylogeographic reconstruction of their homeland impossible (as can be seen in [11]).

Finally, we want to note that even in an expansion scenario, we have to be aware of the opacity of history. The Bantu expansion beautifully illustrates how the first clades inform the reconstruction of the homeland despite a strong directional trend in the later expansion. But what if these first clades were missing or migrated away? In the electronic supplementary material, S8, we show how the reconstruction of the Bantu homeland would fail dramatically in this hypothetical scenario. This emphasizes the importance of complete sampling, especially in early splits and in the regions around a hypothesized homeland. Furthermore, including an outgroup of languages not part of the family under study will stabilize the reconstruction of the homeland. Specifically, [12] included an outgroup of Jarawan and Grassfields languages with the Narrow Bantu languages.

As a further caveat of our results, we want to reiterate the fact that all the geographical reconstructions in the simulation study are informed by the true phylogenetic tree of the simulated languages. In reality, this phylogeny needs to be inferred from linguistic data, which is a difficult and challenging task. Typically, phylogenetic reconstructions are based on models of language evolution through vertical transfer (inheritance), while horizontal transfer of linguistic features (borrowing) would violate the model assumptions. The severity of this problem, i.e. the effect of borrowing on the reconstruction, depends on the type of data used for the analysis—for example, core vocabulary has been proposed to be fairly stable to borrowing [51,52]. Thus, wherever possible it is advisable to remove borrowed features from the data. However, simulation studies have shown that Bayesian phylogenetic reconstructions are resilient to moderate levels of borrowing [21]. For the current study, we intentionally avoided these problems by providing the phylogenetic tree in the reconstruction. Our focus is the geographical model underlying Bayesian phylogeography. Using the true phylogeny allows us to isolate biases in the geographical models from potential errors in the phylogeny, propagating to errors in the geographical reconstruction.

Besides the findings in this study, a phylogeographic reconstruction has to be checked against the historical context. Clearly, including records of ancient languages will improve the reconstruction, but this is often not possible. Instead, therefore, the reconstruction can be corroborated by archaeological or genetic findings. Especially findings of cultural artefacts and practices associated with the language family and knowledge about the spread and development of domesticated crops can give a sense for the plausibility of the reconstruction. More generally, qualitative discussions, as is common in historical linguistics, are an important complement to quantitative, phylogeographic reconstructions. For example, the embedding in a deeper historical context (such as the spatial distribution of other Niger-Congo languages for Bantu), knowledge about contacts with other language families (such as contacts between Indo-European and Uralic languages), or knowledge about trade routes (such as the river-based Arawakan expansion discussed in [53]), as well as detailed studies of reconstructed vocabulary items (as e.g. in [54] for Mayan and [55] for Uto-Aztecan) can be taken into consideration to further support a geographical reconstruction.

# 6. Conclusion

Directional trends in the spatial evolution of languages are a clear case of model misspecification for the most widely used models in phylogeographic analyses. We have demonstrated the effect of directional trends on the reconstruction of language spread in two-movement scenarios—migrations and

expansions—with corresponding historical interpretations. The effects on the reconstructions differ greatly between the two scenarios: migrations (e.g. Chadic and Celtic) make a correct reconstruction impossible, while expansions (e.g. Bantu and Austronesian) only lead to minor imprecisions in the reconstruction, even under severe directional trends. The message to researchers applying phylogeographic methods accordingly depends on the historical scenarios they are investigating: in scenarios of continuous expansions, where a potential directional trend was the result of geographical constraints, our findings support the validity of phylogeographic analysis. In cases where external changes, such as environmental factors or replacement by other languages, might have caused a displacement to the currently inhabited areas, our results strongly discourage the use of phylogeographic reconstruction methods.

Data accessibility. Data and relevant code for this research work are stored in GitHub: https://github.com/NicoNeureiter/drifting_into_nowhere/ and have been archived within the Zenodo repository: https://doi.org/10.5281/zenodo.4279082.

Authors' contributions. N.N. and P.R. designed the research; N.N. implemented the simulations and performed the experiments; B.B. and R.v.G. framed the research in a linguistic and historical context; R.W. coordinated the study; N.N. and P.R. wrote the paper with contributions from R.v.G., B.B. and R.W. All authors gave final approval for publication and agree to be held accountable for the work performed therein.

Competing interests. We declare we have no competing interests.

Funding. This research was supported by the University of Zurich Research Priority Program 'Language and Space', the Swiss NSF Sinergia grant no. CRSII1_160739 'Linguistic Morphology in Time and Space' (LiMiTS), the European Research Council, ERC Consolidator grant no. 818854 'South American Population History Revisited' (SAPPHIRE) and the NCCR Evolving Language, Swiss NSF Agreement No. 51NF40_180888.

Acknowledgements. We thank Curdin Derungs, Péter Jeszenszky and Nour Efrat-Kowalsky for their valuable input, discussions and feedback.

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
