## [Reviewer comments · Royal Society Open Science]

Review History

RSOS-201079.R0 (Original submission)

Review form: Reviewer 1

Is the manuscript scientifically sound in its present form?

Yes

Are the interpretations and conclusions justified by the results?

Yes

Is the language acceptable?

Yes

Do you have any ethical concerns with this paper?

No

Have you any concerns about statistical analyses in this paper?

No

Recommendation?

Accept with minor revision (please list in comments)

Comments to the Author(s)

I fully enjoyed reading Neureiter et al manuscript entitled "Can Bayesian phylogeography reconstruct migrations and expansions in human history?" The authors simulate expansions and migrations of language families and test the accuracy of Bayesian phylogeography methods to reconstruct the homeland of language families from the simulated tree and location of tip leaves. The authors find that expansions, but not migrations, are reconstructed faithfully. They further provide three descriptive statistics to help researchers determine whether the historical expansion of a language family likely follows a migration or expansion scenario, and hence, whether using Bayesian phylogeography can accurately inform about the homeland of the languages.

I believe this paper is a valuable addition to the field of anthropology and cultural evolution, and it will be of interest to linguists, anthropologists, and evolutionary biologists working on cultural matters, as it provides a welcomed insight into the assumptions, limitations, and also potential for good results when reconstructing the spatial evolution of languages.

Abstract

The abstract is clear and got my interest in reading the paper further and finding out more details about the findings. A minor comment would be that it would be helpful to explain what a "grid-based region-growing process" refers to when introducing the expansion. Also, I think the sentence: "Our results show that phylogeography fails to reconstruct migrations" is perhaps too strong. Replacing phylogeography with "current/most widely used/popular/standard/Bayesian phylogeographic methods" is more accurate.

I found the Introduction very well written and easy to follow. Some minor comments:

Page 3 of 14:

- o Lines 33:35: I agree that comprehensive reviews are lacking, as well as extensive discussions on assumptions and biases. However, detailed explanations of how the models work, as well as tutorials can be found in the papers introducing the methods (e.g. reference 11 and 13 in text), or, in the case of the BEAST for example, on the software online page. Worth citing these sources in line 34/35.
- o Line 48: and generally when working with empirical data.
- o Line 54: same comment as the abstract, while the methods will be explained in detail later on, I find the term "grid-based region-growing process" a bit abstract, and it would be helpful to have a short explanation here (akin to page 5/14, lines 18-19).

Page 4 of 14

- o Lines 50-54: this makes sense, however, it would be useful to acknowledge that there are languages with huge (and sometimes scarcely populated) ranges.

Section 2. Phylogeographic modelling - very informative and clear.

Section 3. Methods - I think this section would benefit from a more detailed explanation of the two simulated scenarios. While these scenarios are described at length in the supplement, I think they are essentially the meat of the paper, and readers would benefit from the addition of as much text as possible from Sections S1.1 and S1.2. I do understand though trying to keep the word count under check. Maybe the authors could cut some of the text that explains the gist of these methods? (e.g. while I loved the Phylogeographic modelling section, the knowledge is out there and explained in papers introducing these methods; I would be more interested for more details on the simulations in the current paper). Also, I think a clear explanation of all simulation

scenarios should be included: e.g. for the expansion scenario, the authors simulated under the following degrees of geo constraints: [...]. Regarding this: I take it that the results are summarised over all simulations with varying parameters for the two models. I think this is fine, but I would like to know whether the authors observed any differences between data simulated under different parameter values – e.g. in figures 2 and 3, it would be nice to see (in the supplement) these figures split for specific settings of trend or sector angle.

Migration scenario (Section S1.1):

- o I think the parameters of the model should be explained in the main text: direction and strength of bias, total time span of the simulation, expected number of leaves, and expected distanced covered. Equation (1) is also important to understand how these parameters are incorporated in the random walk, but I understand putting equations in the main text requires considerable addition of text. Parameters of the birth-death process are important to understand the diversification of languages. Importantly too, the main text should have a clear explanation of simulation cases, e.g. after reading the supplement, I thought the authors simulated under three values for the direction & strength of bias: 0, expected distance covered, and 2* expected distance covered, but looking at Figures 2 and 3, I can see that a variety of directional trend values exist? I am not sure I am interpreting this correctly, but in any case, this would benefit from a clear explanation in the main text.

Expansion scenario (Section S1.2):

- o The geographic constraints should be clearly explained in the main text: I particularly found the last paragraph of the Section S1.2 to be very informative. Also, did the authors perform simulations under different degrees of geo-constraints, and if so, which are these? If that is not the case, what is the geo scenario all simulations fall under?

- o I think the parameters of the ExpSim scenario should be mentioned in the main text: pgrow, and the split-size choosing algorithm. Regarding the latter, I was wondering how does the area of one language (70-100cells) compare to the total area occupied by all languages at the end of an expansion scenario, e.g. in general, one language range size represented on average around x% of total perimeter covered by languages at the end of one simulated expansion scenario. This can be mentioned in the supplement, but I think it helps to understand what a range size of 70-100 cells means in the simulated world.

- o Regarding pgrow: is this drawn from a random distribution? What are the parameters of this distribution?

- o Regarding the split of language ranges: how are the perimeters of the two new languages determined? i.e. how is the ancestral range divided into two languages?

- o How does the distribution for the number of time steps for expansion simulations look like?

I think it is also important to mention in the main text that the input tree in the models is the true (simulated) tree, just because, as mentioned in the supplement, these models reconstruct the tree too, so it helps to understand that here the geographic reconstruction is the only focus.

I would personally expand the explanation for RMSE in the main text (e.g. the Euclidean distance between every location in the posterior distribution and the true root). Speaking of the posterior distribution, I would be interested to know how many iterations these runs were set to, and how much was discarded as burn-in. These details can of course go in the supplement, alongside the details about the priors (end of section S2).

I would also expand the explanation of the observed trend in the main text; the explanation in the supplement (“the distance between the homeland of an expansion and the mean of the final locations” seems more straightforward to understand than the “mean displacement” in my opinion).

Results:

Page 7 of 14:

- o Lines 34-37: I got confused about one aspect: do the authors use a relaxed random walk for the two directional trends? (CDRW and RDRW) - in the supplement both the CDRW and RDRW are mentioned, but in the results section, only the CDRW is mentioned.
- o Line 43: regarding historical data, I think it's important to mention how this is added into the migration scenario in the main text (this would also clarify why the expansion scenario only has a "no historical data" panel in figure 3).

I enjoyed reading the Discussion, section (a) is very clear and explains important differences between the expansion and migration scenarios, while section (b) puts these theoretical considerations into practice with a welcomed example. My personal preference would be that the three descriptive statistics in section (a) would be explained in more detail, however, in the interest of saving space, I completely understand why their equations/algorithms have been moved to the supplement.

Also, I wondered if the authors can comment on the following: the descriptive statistics make sense intuitively, and they hold true when computed over the "true" tree, i.e. a tree resulted from simulating a migration or expansion process; however, in an empirical analysis, the tree would be reconstructed alongside the homeland; so I was wondering: would these statistics be the same when the tree is also an output of the model? E.g. would the outputted tree be balanced in a migration scenario, or would the model wrongly output an imbalanced tree?

Review form: Reviewer 2

Is the manuscript scientifically sound in its present form?

Yes

Are the interpretations and conclusions justified by the results?

Yes

Is the language acceptable?

Yes

Do you have any ethical concerns with this paper?

No

Have you any concerns about statistical analyses in this paper?

No

Recommendation?

Accept with minor revision (please list in comments)

Comments to the Author(s)

This paper is a valuable and well-executed contribution to the growing literature on the application of Bayesian phylogenetic methods to linguistic data. Work like this probing the suitability and reliability of widely used models for historical linguistic purposes is sorely needed. I recommend the paper be accepted for publication. Below I include some remarks on how I think the paper could be improved, but I do not consider any of them to be addressing

serious flaws or shortcomings and do not feel that publication should be contingent upon these changes being made.

The introduction/overview of phylogeography disregards the existence of discrete phylogeographic methods. This is perhaps understandable/acceptable in the context of the present paper, as only continuous phylogeographic methods have been used thus far in historical linguistic contexts, but it may still be worthwhile clarifying that phylogeography as a field is a little broader than this. Discrete methods are widely used and influential in biogeography, and function especially well in the context of island groups, where species/languages can be modelled as being either present or absent on each individual island. There's an epidemiological example in the Lemey et al paper cited below - not necessarily the best example citation on discrete phylogeography by any means, just one I happened to be immediately aware of.

Lemey P, Rambaut A, Drummond AJ, Suchard MA (2009) Bayesian Phylogeography Finds Its Roots. *PLoS Computational Biology* 5(9): e1000520.
<https://doi.org/10.1371/journal.pcbi.1000520>

The ExpSim methodology bears some degree of similarity to the the language expansion model described in an earlier paper by Gavin et al (full citation below), and I think it would be appropriate to mention this in the paper and cite that work. I don't doubt for a second that ExpSim was developed independently and don't mean to imply that the authors of that earlier work deserve any credit for the present work - instead I'm just very aware that readers who are interested in using or adapting one model are highly likely to be interested in doing the same to the other, and it's a useful secondary purpose of scientific papers to help "knit the literature together" and enable this kind of idea discovery.

Gavin, M.C., Rangel, T.F., Bowern, C., Colwell, R.K., Kirby, K.R., Botero, C.A., Dunn, M., Dunn, R.R., McCarter, J., Pacheco Coelho, M.T. and Gray, R.D. (2017), Process-based modelling shows how climate and demography shape language diversity. *Global Ecol. Biogeogr.*, 26: 584-591.
[doi:10.1111/geb.12563](https://doi.org/10.1111/geb.12563)

It was not immediately clear to me on my first reading what the "HPD coverage" quantity shown in Figures 2b and 3b actually was, although reading the relevant paragraph on page S4 of the Supporting Information clarified this. I would suggest that perhaps some of this wording be migrated from the SI to the main paper. Even after reading the SI, though, I'm not actually sure exactly how the "HPD region" is defined. I guess it's the convex hull of all the sampled root locations?

I remain a little unclear on a few mechanical details about the ExpSim model, which aren't really essential to appreciating its role in this paper but which nevertheless it would nice to have explained so readers don't have to resort to reading the code. How are fossil locations generated for languages under this model, when the languages themselves occupy a region of space (like real languages!) while continuous phylogeographic models model language locations as individual points (an unfortunate simplification)? When a language grows to its randomly sampled splitting size, how does the split actually happen? Is the new language created as a single cell neighbouring the split language? Or are half of the cells in the splitting language reassigned to the new language?

The authors are to be commended on making all the code required for a full reproduction of the present study freely available in a public GitHub repository. I would strongly encourage the authors to add an explicit license to the code so that it is unambiguous that other researchers can adapt it for future use. In principle, this code should also be subject to peer review and I'd have loved to do this myself but time pressures prevented it. I would encourage the authors to

consider the prospect of releasing the MigSim and ExpSim tools as separate packages, independent of the exact analyses in this paper, with generalised interfaces and user documentation. Such a tool could be submitted to a software-oriented journal, such as Journal of Open Research Software, to give it a better chance to be scrutinised.

Minor comments on the writing:

In the first paragraph to begin on page 3, there is a missing author name for reference 18 (it appears currently as "author?" in bold).

The term "demic process" should be defined immediately after its first use.

It's a very small thing to point out, but my earlier academic background has left me a bit of a stickler for correct use of mathematical notation, so I feel compelled (sorry!) to do a little nit-picking regarding section S4.1 of the Support Information. When describing the range of birth- and death-rates used in MigSim, the authors make use of the set membership symbol, but do not use the standard "curly brackets" to denote the set of values. They use "square brackets" - with two values separated by commas between them, these would denote a closed interval, which is indeed a set and hence valid on the right hand side of a set membership symbol, but with three numbers inbetween this just reads as gibberish to a mathematically literate reader. In a similarly pedantic vein, the authors say they varied the split size between (140,200) and (25,33). This notation denotes open intervals, i.e. the set of all real numbers between, but not including, the endpoints. I suspect the endpoints actually were used in this case (though I could be wrong), however even closed intervals of [140,200] and [25,33] wouldn't be appropriate here because MigSim's split size can only take on integer values. It's probably simpler to just use words here, as indeed is done when describing the default case where split size is sampled between 70 and 100.

Review form: Reviewer 3

Is the manuscript scientifically sound in its present form?

No

Are the interpretations and conclusions justified by the results?

Yes

Is the language acceptable?

Yes

Do you have any ethical concerns with this paper?

No

Have you any concerns about statistical analyses in this paper?

No

Recommendation?

Major revision is needed (please make suggestions in comments)

Comments to the Author(s)

This paper aims to study the relevance of phylogeographic methods for reconstructing spatiotemporal patterns of language diversification. The originality of the study is to use

simulations of language diversification to test the validity of phylogeographic approaches. The authors simulate expansions and long-distance migrations of populations (languages?) and demonstrate that expansions can be relatively efficiently traced, but migrations cannot be traced at all, especially if no historical data are available. The authors conclude with recommendations on the use of phylogeographic methods to reconstruct the spatial evolution of languages.

The paper is well written, and the statistical analyses seem correct as far as I can judge. I think that the results of this paper may be potentially interesting, but I see several major problems in its present state, that must be addressed prior to publication.

There is for me a confusion throughout the paper between biological (or genetic) evolution and linguistic evolution. While the two may be correlated in some cases, they can also be completely disconnected when cultural traits are transmitted between populations without involving movements of populations. While this issue is briefly introduced, it should be discussed in much greater depth both in the introduction and in the discussion, especially as the authors mention that this is a crucial point in the discussion of the results (page 4 line 41). This confusion can also be seen in the terms used by the authors, for example, the term phylogenetic is used many times when there is no analysis or simulation of genetic data in this study. I can quote in the supplementary material the phrase "While diversification is important to simulate phylogenetically related languages". It is not clear to me what "phylogenetically related languages" means. Furthermore, the title states that it is about "migrations and expansions in human history" whereas it is about migrations and expansion of LANGUAGES. It therefore seems to me essential that the authors clarify throughout the paper whether they are talking about simulations of languages or populations and that they are more consistent in the terms used.

This brings me to my second major point. The originality of this study lies in the simulation of the evolution of languages during an expansion or a migration of populations, but I am unable to assess whether the chosen model of dispersion/diversification is a good representation of this process. I find that the method section of the paper is far too short concerning simulations and that they should be explained in more detail. To my view, part of the supplementary material should be included in the main text. I think it is necessary to better explain the assumptions underlying this model from the point of view of linguistic evolution and if possible, to compare the model developed with other existing models. Being unfamiliar with this literature, I lacked elements to assess the realism of the model proposed to simulate the evolution of language, but it seems odd to me that languages evolve only through dispersal and split. A better justification of the modelling approach is thus necessary. The articles cited in the present manuscript that use phylogeographic approaches to reconstruct language evolution have been criticized precisely on the ground that language evolution cannot be reconstructed in the same way as the evolution of genetic diversity. It seems necessary to me that the authors of the study position themselves in relation to this hypothesis and place their model in a wider context of simulating linguistic evolution.

I think that once these major issues have been clarified, it will be easier to evaluate the importance of the findings.

Decision letter (RSOS-201079.R0)

Dear Mr Neureiter

The Editors assigned to your paper RSOS-201079 "Can Bayesian phylogeography reconstruct migrations and expansions in human history?" have now received comments from reviewers and would like you to revise the paper in accordance with the reviewer comments and any comments from the Editors. Please note this decision does not guarantee eventual acceptance.

Please submit your revised manuscript and required files (see below) no later than 21 days from today's (ie 20-Oct-2020) date. Note: the ScholarOne system will 'lock' if submission of the revision is attempted 21 or more days after the deadline. If you do not think you will be able to meet this deadline please contact the editorial office immediately.

on behalf of Dr Dieter Lukas (Associate Editor) and Kevin Padian (Subject Editor)
openscience@royalsociety.org

Associate Editor Comments to Author (Dr Dieter Lukas):

Dear authors,

I agree with the reviewers that your article is an important contribution. However, I also agree that revisions are necessary to indicate the specific contribution within the larger existing field. In particular, I agree that it would help to include more details on the models in the manuscript. I also agree with reviewer 3 that it is necessary to discuss the potential limitations in modelling the transmission of a cultural trait such as language as a genetically inherited trait, rather than taking other transmission modes into account (for example, horizontal transmission is particularly likely to happen among societies in close geographic proximity, which could have a large influence on

the inferred patterns – e.g. see <https://doi.org/10.1098/rstb.2017.0054>). Given that there are no length restrictions at RSOS, I encourage the authors to expand the relevant sections to clarify these relevant points.

Sincerely,
Dieter Lukas
(Associate Editor)

Reviewer comments to Author:

Reviewer: 1

Comments to the Author(s)

I fully enjoyed reading Neureiter et al manuscript entitled “Can Bayesian phylogeography reconstruct migrations and expansions in human history?” The authors simulate expansions and migrations of language families and test the accuracy of Bayesian phylogeography methods to reconstruct the homeland of language families from the simulated tree and location of tip leaves. The authors find that expansions, but not migrations, are reconstructed faithfully. They further provide three descriptive statistics to help researchers determine whether the historical expansion of a language family likely follows a migration or expansion scenario, and hence, whether using Bayesian phylogeography can accurately inform about the homeland of the languages.

I believe this paper is a valuable addition to the field of anthropology and cultural evolution, and it will be of interest to linguists, anthropologists, and evolutionary biologists working on cultural matters, as it provides a welcomed insight into the assumptions, limitations, and also potential for good results when reconstructing the spatial evolution of languages.

Abstract

The abstract is clear and got my interest in reading the paper further and finding out more details about the findings. A minor comment would be that it would be helpful to explain what a “grid-based region-growing process” refers to when introducing the expansion. Also, I think the sentence: “Our results show that phylogeography fails to reconstruct migrations” is perhaps too strong. Replacing phylogeography with “current/most widely used/popular/standard/Bayesian phylogeographic methods” is more accurate.

I found the Introduction very well written and easy to follow. Some minor comments:

Page 3 of 14:

- o Lines 33:35: I agree that comprehensive reviews are lacking, as well as extensive discussions on assumptions and biases. However, detailed explanations of how the models work, as well as tutorials can be found in the papers introducing the methods (e.g. reference 11 and 13 in text), or, in the case of the BEAST for example, on the software online page. Worth citing these sources in line 34/35.

- o Line 48: and generally when working with empirical data.

- o Line 54: same comment as the abstract, while the methods will be explained in detail later on, I find the term “grid-based region-growing process” a bit abstract, and it would be helpful to have a short explanation here (akin to page 5/14, lines 18-19).

Page 4 of 14

- o Lines 50-54: this makes sense, however, it would be useful to acknowledge that there are languages with huge (and sometimes scarcely populated) ranges.

Section 2. Phylogeographic modelling - very informative and clear.

Section 3. Methods – I think this section would benefit from a more detailed explanation of the two simulated scenarios. While these scenarios are described at length in the supplement, I think they are essentially the meat of the paper, and readers would benefit from the addition of as much text as possible from Sections S1.1 and S1.2. I do understand though trying to keep the word count under check. Maybe the authors could cut some of the text that explains the gist of these methods? (e.g. while I loved the Phylogeographic modelling section, the knowledge is out there and explained in papers introducing these methods; I would be more interested for more details on the simulations in the current paper). Also, I think a clear explanation of all simulation scenarios should be included: e.g. for the expansion scenario, the authors simulated under the following degrees of geo constraints: [...]. Regarding this: I take it that the results are summarised over all simulations with varying parameters for the two models. I think this is fine, but I would like to know whether the authors observed any differences between data simulated under different parameter values – e.g. in figures 2 and 3, it would be nice to see (in the supplement) these figures split for specific settings of trend or sector angle.

Migration scenario (Section S1.1):

o I think the parameters of the model should be explained in the main text: direction and strength of bias, total time span of the simulation, expected number of leaves, and expected distanced covered. Equation (1) is also important to understand how these parameters are incorporated in the random walk, but I understand putting equations in the main text requires considerable addition of text. Parameters of the birth-death process are important to understand the diversification of languages. Importantly too, the main text should have a clear explanation of simulation cases, e.g. after reading the supplement, I thought the authors simulated under three values for the direction & strength of bias: 0, expected distance covered, and 2* expected distance covered, but looking at Figures 2 and 3, I can see that a variety of directional trend values exist? I am not sure I am interpreting this correctly, but in any case, this would benefit from a clear explanation in the main text.

Expansion scenario (Section S1.2):

o The geographic constraints should be clearly explained in the main text: I particularly found the last paragraph of the Section S1.2 to be very informative. Also, did the authors perform simulations under different degrees of geo-constraints, and if so, which are these? If that is not the case, what is the geo scenario all simulations fall under?

o I think the parameters of the ExpSim scenario should be mentioned in the main text: p_{grow}, and the split-size choosing algorithm. Regarding the latter, I was wondering how does the area of one language (70-100cells) compare to the total area occupied by all languages at the end of an expansion scenario, e.g. in general, one language range size represented on average around x% of total perimeter covered by languages at the end of one simulated expansion scenario. This can be mentioned in the supplement, but I think it helps to understand what a range size of 70-100 cells means in the simulated world.

o Regarding p_{grow}: is this drawn from a random distribution? What are the parameters of this distribution?

o Regarding the split of language ranges: how are the perimeters of the two new languages determined? i.e. how is the ancestral range divided into two languages?

o How does the distribution for the number of time steps for expansion simulations look like?

I think it is also important to mention in the main text that the input tree in the models is the true (simulated) tree, just because, as mentioned in the supplement, these models reconstruct the tree too, so it helps to understand that here the geographic reconstruction is the only focus.

I would personally expand the explanation for RMSE in the main text (e.g. the Euclidean distance between every location in the posterior distribution and the true root). Speaking of the posterior distribution, I would be interested to know how many iterations these runs were set to, and how

much was discarded as burn-in. These details can of course go in the supplement, alongside the details about the priors (end of section S2).

I would also expand the explanation of the observed trend in the main text; the explanation in the supplement (“the distance between the homeland of an expansion and the mean of the final locations” seems more straightforward to understand than the “mean displacement” in my opinion).

Results:

Page 7 of 14:

- o Lines 34-37: I got confused about one aspect: do the authors use a relaxed random walk for the two directional trends? (CDRW and RDRW) - in the supplement both the CDRW and RDRW are mentioned, but in the results section, only the CDRW is mentioned.
- o Line 43: regarding historical data, I think it's important to mention how this is added into the migration scenario in the main text (this would also clarify why the expansion scenario only has a “no historical data” panel in figure 3).

I enjoyed reading the Discussion, section (a) is very clear and explains important differences between the expansion and migration scenarios, while section (b) puts these theoretical considerations into practice with a welcomed example. My personal preference would be that the three descriptive statistics in section (a) would be explained in more detail, however, in the interest of saving space, I completely understand why their equations/algorithms have been moved to the supplement.

Also, I wondered if the authors can comment on the following: the descriptive statistics make sense intuitively, and they hold true when computed over the “true” tree, i.e. a tree resulted from simulating a migration or expansion process; however, in an empirical analysis, the tree would be reconstructed alongside the homeland; so I was wondering: would these statistics be the same when the tree is also an output of the model? E.g. would the outputted tree be balanced in a migration scenario, or would the model wrongly output an imbalanced tree?

Reviewer: 2

Comments to the Author(s)

This paper is a valuable and well-executed contribution to the growing literature on the application of Bayesian phylogenetic methods to linguistic data. Work like this probing the suitability and reliability of widely used models for historical linguistic purposes is sorely needed. I recommend the paper be accepted for publication. Below I include some remarks on how I think the paper could be improved, but I do not consider any of them to be addressing serious flaws or shortcomings and do not feel that publication should be contingent upon these changes being made.

The introduction/overview of phylogeography disregards the existence of discrete phylogeographic methods. This is perhaps understandable/acceptable in the context of the present paper, as only continuous phylogeographic methods have been used thus far in historical linguistic contexts, but it may still be worthwhile clarifying that phylogeography as a field is a little broader than this. Discrete methods are widely used and influential in biogeography, and function especially well in the context of island groups, where species/languages can be modelled as being either present or absent on each individual island. There's an epidemiological example in the Lemey et al paper cited below - not necessarily the best example citation on discrete phylogeography by any means, just one I happened to be immediately aware of.

Lemey P, Rambaut A, Drummond AJ, Suchard MA (2009) Bayesian Phylogeography Finds Its Roots. *PLoS Computational Biology* 5(9): e1000520.
<https://doi.org/10.1371/journal.pcbi.1000520>

The ExpSim methodology bears some degree of similarity to the the language expansion model described in an earlier paper by Gavin et al (full citation below), and I think it would be appropriate to mention this in the paper and cite that work. I don't doubt for a second that ExpSim was developed independently and don't mean to imply that the authors of that earlier work deserve any credit for the present work - instead I'm just very aware that readers who are interested in using or adapting one model are highly likely to be interested in doing the same to the other, and it's a useful secondary purpose of scientific papers to help "knit the literature together" and enable this kind of idea discovery.

Gavin, M.C., Rangel, T.F., Bowern, C., Colwell, R.K., Kirby, K.R., Botero, C.A., Dunn, M., Dunn, R.R., McCarter, J., Pacheco Coelho, M.T. and Gray, R.D. (2017), Process-based modelling shows how climate and demography shape language diversity. *Global Ecol. Biogeogr.*, 26: 584-591.
[doi:10.1111/geb.12563](https://doi.org/10.1111/geb.12563)

It was not immediately clear to me on my first reading what the "HPD coverage" quantity shown in Figures 2b and 3b actually was, although reading the relevant paragraph on page S4 of the Supporting Information clarified this. I would suggest that perhaps some of this wording be migrated from the SI to the main paper. Even after reading the SI, though, I'm not actually sure exactly how the "HPD region" is defined. I guess it's the convex hull of all the sampled root locations?

I remain a little unclear on a few mechanical details about the ExpSim model, which aren't really essential to appreciating its role in this paper but which nevertheless it would nice to have explained so readers don't have to resort to reading the code. How are fossil locations generated for languages under this model, when the languages themselves occupy a region of space (like real languages!) while continuous phylogeographic models model language locations as individual points (an unfortunate simplification)? When a language grows to its randomly sampled splitting size, how does the split actually happen? Is the new language created as a single cell neighbouring the split language? Or are half of the cells in the splitting language reassigned to the new language?

The authors are to be commended on making all the code required for a full reproduction of the present study freely available in a public GitHub repository. I would strongly encourage the authors to add an explicit license to the code so that it is unambiguous that other researchers can adapt it for future use. In principle, this code should also be subject to peer review and I'd have loved to do this myself but time pressures prevented it. I would encourage the authors to consider the prospect of releasing the MigSim and ExpSim tools as separate packages, independent of the exact analyses in this paper, with generalised interfaces and user documentation. Such a tool could be submitted to a software-oriented journal, such as *Journal of Open Research Software*, to give it a better chance to be scrutinised.

Minor comments on the writing:

In the first paragraph to begin on page 3, there is a missing author name for reference 18 (it appears currently as "author?" in bold).

The term "demic process" should be defined immediately after its first use.

It's a very small thing to point out, but my earlier academic background has left me a bit of a stickler for correct use of mathematical notation, so I feel compelled (sorry!) to do a little nit-picking regarding section S4.1 of the Support Information. When describing the range of birth- and death-rates used in MigSim, the authors make use of the set membership symbol, but do not use the standard "curly brackets" to denote the set of values. They use "square brackets" - with two values separated by commas between them, these would denote a closed interval, which is indeed a set and hence valid on the right hand side of a set membership symbol, but with three numbers inbetween this just reads as gibberish to a mathematically literate reader. In a similarly pedantic vein, the authors say they varied the split size between (140,200) and (25,33). This notation denotes open intervals, i.e. the set of all real numbers between, but not including, the endpoints. I suspect the endpoints actually were used in this case (though I could be wrong), however even closed intervals of [140,200] and [25,33] wouldn't be appropriate here because MigSim's split size can only take on integer values. It's probably simpler to just use words here, as indeed is done when describing the default case where split size is sampled between 70 and 100.

Reviewer: 3

Comments to the Author(s)

This paper aims to study the relevance of phylogeographic methods for reconstructing spatiotemporal patterns of language diversification. The originality of the study is to use simulations of language diversification to test the validity of phylogeographic approaches. The authors simulate expansions and long-distance migrations of populations (languages?) and demonstrate that expansions can be relatively efficiently traced, but migrations cannot be traced at all, especially if no historical data are available. The authors conclude with recommendations on the use of phylogeographic methods to reconstruct the spatial evolution of languages.

The paper is well written, and the statistical analyses seem correct as far as I can judge. I think that the results of this paper may be potentially interesting, but I see several major problems in its present state, that must be addressed prior to publication.

There is for me a confusion throughout the paper between biological (or genetic) evolution and linguistic evolution. While the two may be correlated in some cases, they can also be completely disconnected when cultural traits are transmitted between populations without involving movements of populations. While this issue is briefly introduced, it should be discussed in much greater depth both in the introduction and in the discussion, especially as the authors mention that this is a crucial point in the discussion of the results (page 4 line 41). This confusion can also be seen in the terms used by the authors, for example, the term phylogenetic is used many times when there is no analysis or simulation of genetic data in this study. I can quote in the supplementary material the phrase "While diversification is important to simulate phylogenetically related languages". It is not clear to me what "phylogenetically related languages" means. Furthermore, the title states that it is about "migrations and expansions in human history" whereas it is about migrations and expansion of LANGUAGES. It therefore seems to me essential that the authors clarify throughout the paper whether they are talking about simulations of languages or populations and that they are more consistent in the terms used.

This brings me to my second major point. The originality of this study lies in the simulation of the evolution of languages during an expansion or a migration of populations, but I am unable to assess whether the chosen model of dispersion/diversification is a good representation of this process. I find that the method section of the paper is far too short concerning simulations and that they should be explained in more detail. To my view, part of the supplementary material should be included in the main text. I think it is necessary to better explain the assumptions underlying this model from the point of view of linguistic evolution and if possible, to compare

the model developed with other existing models. Being unfamiliar with this literature, I lacked elements to assess the realism of the model proposed to simulate the evolution of language, but it seems odd to me that languages evolve only through dispersal and split. A better justification of the modelling approach is thus necessary. The articles cited in the present manuscript that use phylogeographic approaches to reconstruct language evolution have been criticized precisely on the ground that language evolution cannot be reconstructed in the same way as the evolution of genetic diversity. It seems necessary to me that the authors of the study position themselves in relation to this hypothesis and place their model in a wider context of simulating linguistic evolution.

I think that once these major issues have been clarified, it will be easier to evaluate the importance of the findings.

===PREPARING YOUR MANUSCRIPT===

===PREPARING YOUR REVISION IN SCHOLARONE===

Author's Response to Decision Letter for (RSOS-201079.R0)

See Appendix A.

Decision letter (RSOS-201079.R1)

Dear Mr Neureiter

On behalf of the Editors, we are pleased to inform you that your Manuscript RSOS-201079.R1 "Can Bayesian phylogeography reconstruct migrations and expansions in linguistic evolution?" has been accepted for publication in Royal Society Open Science subject to minor revision in accordance with the referees' reports. Please find the referees' comments along with any feedback from the Editors below my signature.

We invite you to respond to the comments and revise your manuscript. Below the Editor's comments (where applicable) we provide additional requirements. Final acceptance of your manuscript is dependent on these requirements being met. We provide guidance below to help you prepare your revision.

Please submit your revised manuscript and required files (see below) no later than 7 days from today's (ie 30-Nov-2020) date. Note: the ScholarOne system will 'lock' if submission of the revision is attempted 7 or more days after the deadline. If you do not think you will be able to meet this deadline please contact the editorial office immediately.

on behalf of Dr Dieter Lukas (Associate Editor) and Kevin Padian (Subject Editor)
openscience@royalsociety.org

Associate Editor Comments to Author (Dr Dieter Lukas):

Associate Editor

Comments to the Author:

The authors have done a great job in addressing the reviewers' comments. The revised manuscript now contains the details necessary to understand the approach and explains the assumptions and limitations.

I only have a few suggestions to potentially change some wording in the introduction to help readers who might not be as familiar with phylogeographic approaches:

- In Line 101ff you state: "In this article we focus on the geographic component of phylogeographic models and assume a setting where the phylogenetic tree is faithfully reconstructed."

I think it might help if in the introduction you add a brief explanation of how your approach links to empiricists: it seems to be that there exists a dataset of language that diversified from each other for which you have an inferred tree reflecting which languages are related and when they split, and for each language you also know their current location (either as a centre point or as their range) - based on this the question is can we infer where speakers of the ancestral form of the language were located. I think making this more explicit will also help readers who are not familiar with simulations to understand your approach of mechanistically reflecting the processes that might generate the data.

- In Line 123f you talk about: "accuracy of the phylogeographic reconstruction" and in Line 192 you state: "We evaluate the performance of phylogeographic reconstructions in a simulation study."

The introduction never clarifies what the actual focus is and how you are going to assess accuracy. You could assess whether the location of the common ancestor is placed in roughly the right place, you could assess the false positive and false negative rate of detecting directional trends, you could see whether a migration scenario can be differentiated from an expansion scenario, etc. It would help to specify in the introduction what the key outcome result is that you will focus on and that you will use to assess accuracy. It's present in lines 306ff, but it would help the reader understand how this study relates to their work and the questions they are interested in. This could link again to specify what data exactly you generate in the simulations (see previous comment).

- In Line 106ff you state "We implement two simulation scenarios"

It looks like there are three simulations in total: migration with point locations, expansion with point location, and expansion with geographic areas. Maybe start with "We first implement"

- In Line 92 you state: "languages, which disperse through both population movements and through language shifts. Which of these actually was the driver in a specific setting can only be a matter of interpretation in phylogeographic studies. As such, it is not central to this article"

This seems to assume that the rate of change and distance that can be covered are the same for the two processes. I guess there could be additional noise: for example, movement of populations might be covering larger distances whereas language displacement presumably would involve direct contact. That is, population movements might be more like the migration model you simulate, whereas language shifts might be more like the expansion model?

-In the Legend of Figure 5, as pointed out previously by a reviewer, maybe briefly explain what HPD coverage means (in the legend you use the term credible regions rather than HPD coverage).

- I have not gone through the whole manuscript in detail but you might want to check for small remaining grammatical errors, eg. Line 122 remove the comma in "migration and expansion processes, lead to quite"

===PREPARING YOUR MANUSCRIPT===

===PREPARING YOUR REVISION IN SCHOLARONE===

- If you are requesting a discretionary waiver for the article processing charge, the waiver form must be included at this step.
- If you are providing image files for potential cover images, please upload these at this step, and inform the editorial office you have done so. You must hold the copyright to any image provided.
- A copy of your point-by-point response to referees and Editors. This will expedite the preparation of your proof.

- Ensure that your data access statement meets the requirements at <https://royalsociety.org/journals/authors/author-guidelines/#data>. You should ensure that you cite the dataset in your reference list. If you have deposited data etc in the Dryad repository, please only include the 'For publication' link at this stage. You should remove the 'For review' link.
- If you are requesting an article processing charge waiver, you must select the relevant waiver option (if requesting a discretionary waiver, the form should have been uploaded at Step 3 'File upload' above).
- If you have uploaded ESM files, please ensure you follow the guidance at <https://royalsociety.org/journals/authors/author-guidelines/#supplementary-material> to include a suitable title and informative caption. An example of appropriate titling and captioning may be found at https://figshare.com/articles/Table_S2_from_Is_there_a_trade-off_between_peak_performance_and_performance_breadth_across_temperatures_for_aerobic_scope_in_teleost_fishes_/3843624.

Author's Response to Decision Letter for (RSOS-201079.R1)

See Appendix B.

Decision letter (RSOS-201079.R2)

Dear Mr Neureiter,

It is a pleasure to accept your manuscript entitled "Can Bayesian phylogeography reconstruct migrations and expansions in linguistic evolution?" in its current form for publication in Royal Society Open Science.

You can expect to receive a proof of your article in the near future. Please contact the editorial office (openscience_proofs@royalsociety.org) and the production office (openscience@royalsociety.org) to let us know if you are likely to be away from e-mail contact -- if

you are going to be away, please nominate a co-author (if available) to manage the proofing process, and ensure they are copied into your email to the journal.

on behalf of Dr Dieter Lukas (Associate Editor) and Kevin Padian (Subject Editor)
openscience@royalsociety.org

Appendix A

Response to Reviewers

Many thanks to all referees for taking the time to review our paper and for providing such thoughtful and detailed comments. We think these comments have helped us to improve the quality of the paper and to convey the results of our study more clearly. In this document we provide direct responses to the comment of each reviewer (with our answers in **red typeface**). We tried to always quote the relevant changes and we have highlighted the changes in the revised paper in color (in .pdf and .tex format). Line numbers in our responses always refer to the annotated PDF files (EvaluatingPhylogeography_changes.pdf and EvaluatingPhylogeography_SM_changes.pdf).

The major revisions are:

1. As suggested by Reviewer 1, Reviewer 3 and the Editor, we moved the detailed description of the simulation study from the Supplementary Material to the main article.
2. Following suggestions by Reviewer 1 and Reviewer 2, we have added further details on the simulations, on the configuration of the phylogeographic analysis, and on the computation of the HPD coverage to the Supplementary Material.
3. As suggested by Reviewer 1, we added additional figures to the supplement, showing results for the RDRW model and presenting all results in terms of the simulated parameters (trend and sector angle).
4. As suggested by Reviewer 3 and the Editor, we added clarifications on 1) the difference between genetic and linguistic evolution and 2) the assumptions of a tree model of linguistic evolution. In this context, we have also made the terminology we use throughout the paper more consistent.

Reviewer 1

I fully enjoyed reading Neureiter et al manuscript entitled “Can Bayesian phylogeography reconstruct migrations and expansions in human history?” The authors simulate expansions and migrations of language families and test the accuracy of Bayesian phylogeography methods to reconstruct the homeland of language families from the simulated tree and location of tip leaves. The authors find that expansions, but not migrations, are reconstructed faithfully. They further provide three descriptive statistics to help researchers determine whether the historical expansion of a language family likely follows a migration or expansion scenario, and hence, whether using Bayesian phylogeography can accurately inform about the homeland of the languages.

I believe this paper is a valuable addition to the field of anthropology and cultural evolution, and it will be of interest to linguists, anthropologists, and evolutionary biologists working on cultural matters, as it provides a welcomed insight into the assumptions, limitations, and also potential for good results when reconstructing the spatial evolution of languages.

Thank you very much for your detailed feedback. Your comments helped us to hopefully make the paper clearer and more accessible to the reader.

Abstract

The abstract is clear and got my interest in reading the paper further and finding out more details about the findings. A minor comment would be that it would be helpful to explain what a “grid-based region-growing process” refers to when introducing the expansion.

We have realized that the abstract exceeded the RSOS limit of 200 words and thus shortened it. In doing so we dropped the admittedly opaque term “grid-based region-growing process” and now simply state that we “simulate migration and expansion in two scenarios with varying degrees of spatial directional trends”, which is further explained in the Introduction.

Also, I think the sentence: “Our results show that phylogeography fails to reconstruct migrations” is perhaps too strong. Replacing phylogeography with “current/most widely used/popular/standard/Bayesian phylogeographic methods” is more accurate.

We agree that this was overstated. We now refer back to the mention of "state-of-the-art phylogeographic methods" in the previous sentence.

Page 3 of 14:

o Lines 33:35: I agree that comprehensive reviews are lacking, as well as extensive discussions on assumptions and biases. However, detailed explanations of how the models work, as well as tutorials can be found in the papers introducing the methods (e.g. reference 11 and 13 in text), or, in the case of the BEAST for example, on the software online page. Worth citing these sources in line 34/35.

We rephrased the statement to: *"We see a lack of literature discussing the assumptions and biases of Bayesian phylogeography, in particular regarding what kind of spatio-temporal processes can actually be reconstructed."* [lines 28-29]

o Line 48: and generally when working with empirical data.

We added this comment in the text. [lines 43-44]

o Line 54: same comment as the abstract, while the methods will be explained in detail later on, I find the term “grid-based region-growing process” a bit abstract, and it would be helpful to have a short explanation here (akin to page 5/14, lines 18-19).

We added a short explanation of the process: *"[...] grid-based region-growing process where populations or languages occupy cells in a geographic grid and randomly expand to neighbouring cells."* [lines 48-49]

Page 4 of 14

o Lines 50-54: this makes sense, however, it would be useful to acknowledge that there are languages with huge (and sometimes scarcely populated) ranges.

We now mention this in the introduction: *"While in reality there are languages with huge and sometimes scarcely populated ranges, the size constraint generally seems to be a reasonable assumption, since [...]"* [lines 116-117]

Section 2. Phylogeographic modelling - very informative and clear.

Section 3. Methods – I think this section would benefit from a more detailed explanation of the two simulated scenarios. While these scenarios are described at length in the supplement, I think they are essentially the meat of the paper, and readers would benefit from the addition of as much text as possible from Sections S1.1 and S1.2. I do understand though trying to keep the word count under check. Maybe the authors could cut some of the text that explains the gist of these methods? (e.g. while I loved the Phylogeographic modelling section, the knowledge is out there and explained in papers introducing these methods; I would be more interested for more details on the simulations in the current paper). Also, I think a clear explanation of all simulation scenarios should be included: e.g. for the expansion scenario, the authors simulated under the following degrees of geo constraints: [...].

We agree that the Methods section, as it was contained in the Supplementary Material, indeed forms a core part of this article. Our initial reason to place it in the supplement, was indeed to keep the paper short. Since the Editor reminded us that there are no restrictions on the length of the article, we now moved Sections S1-S3 to the main article (now Section 3, replacing the summary text in the old Methods section).

Regarding this: I take it that the results are summarised over all simulations with varying parameters for the two models. I think this is fine, but I would like to know whether the authors observed any differences between data simulated under different parameter values – e.g. in figures 2 and 3, it would be nice to see (in the supplement) these figures split for specific settings of trend or sector angle.

There is almost a 1-to-1 mapping between trend/sector angle and observed trend (only "almost" because of stochasticity). Thus, in Figure 2 and 3 the x-axis (observed trend) implicitly reflects these different settings. We now added plots to the Supplementary Material, which show the results explicitly in terms of the trend and sector angle (Figures S8 and S9). [Supplementary Material, lines 123-133]

Migration scenario (Section S1.1):

o I think the parameters of the model should be explained in the main text: direction and strength of bias, total time span of the simulation, expected number of leaves, and expected distanced covered. Equation (1) is also important to understand how these parameters are incorporated in the random walk, but I understand putting equations in the main text requires considerable addition of text.

We understand that these parameters are relevant for the reader and moved the corresponding explanations from the supplement to the main paper (as part of Section 1.1). [lines 223-238]

Parameters of the birth-death process are important to understand the diversification of languages.

Analogous to the previous comment: we moved the explanations of the birth and death rate to the main paper as part of Section 1.1. [lines 241-247]

Importantly too, the main text should have a clear explanation of simulation cases, e.g. after reading the supplement, I thought the authors simulated under three values for the direction & strength of bias: 0, expected distance covered, and 2* expected distance covered, but looking at Figures 2 and 3, I can see that a variety of directional trend values exist? I am not sure I am interpreting this correctly, but in any case, this would benefit from a clear explanation in the main text.

We added more detail on how we vary the directional trend in both settings in the last paragraph of Section 3 (c). [lines 335-340]

Furthermore, moving S1-S3 to the main paper should help in making the settings clearer.

Added text: "In the MigSim scenario we simulated a total bias μ ranging from 0 to 6000 km (in steps of 500 km) and in the ExpSim scenario the sector angle α ranged from 2π down to 0.2π (in steps of 0.2π). In both scenarios these settings resulted in an observed trend $\hat{\mu}$ between 0 and 6000 km (see Figure 4 and 5)."

Expansion scenario (Section S1.2):

o The geographic constraints should be clearly explained in the main text: I particularly found the last paragraph of the Section S1.2 to be very informative. Also, did the authors perform simulations under different degrees of geo-constraints, and if so, which are these? If that is not the case, what is the geo scenario all simulations fall under?

The sector angle α was varied from 0.2π to 2π (in steps of 0.2π). We think moving S1-S3 and the added explanations to Section 3 (c) should clarify this. [lines 335-340]

o I think the parameters of the ExpSim scenario should be mentioned in the main text: $pgrow$, and the split-size choosing algorithm.

We moved the explanation of these parameters to the main paper as part of Section S1.2 (now 3a). [lines 263-270]

Regarding the latter, I was wondering how does the area of one language (70-100cells) compare to the total area occupied by all languages at the end of an expansion scenario, e.g. in general, one language range size represented on average around x% of total perimeter covered by languages at the end of one simulated expansion scenario. This can be mentioned in the supplement, but I think it helps to understand what a range size of 70-100 cells means in the simulated world.

We added this as a note in the Supplementary Material (S2.2): "The split size of 70 to 100 cells along with the geographic constraints results in extant language with an area of approximately 50 cells on average, equivalent to approximately 1% of the whole simulated area (this was stable across the different cone angle settings)." [Supplementary Material, lines 68-73]

o Regarding $pgrow$: is this drawn from a random distribution? What are the parameters of this distribution?

$pgrow$ is drawn uniformly from the interval $[0,1]$, as we now explain in Section 3a (ii):

"We implement heterogeneity in the growth rate by drawing $pgrow$ uniformly from the interval $[0,1]$ for each language." [lines 265-267]

o Regarding the split of language ranges: how are the perimeters of the two new languages determined? i.e. how is the ancestral range divided into two languages?

We added an explanation of the splitting process to the Supplementary Material (S2.2):

"When a language splits, the cells of the area are divided along a straight line and each daughter language inherits the cells on one side of this line. To ensure that the daughter languages still comprise reasonably compact areas, we split the previous area along the axis of maximum variance (illustrated in Figure S1)." [Supplementary Material lines 74-77]

o How does the distribution for the number of time steps for expansion simulations look like?

Like the MigSim experiments, we also ran the ExpSim experiments for 5000 steps. We added this in Section 3a (ii). [line 271]

I think it is also important to mention in the main text that the input tree in the models is the true (simulated) tree, just because, as mentioned in the supplement, these models reconstruct the tree too, so it helps to understand that here the geographic reconstruction is the only focus.

We moved the explanation to the main paper as part of Section S2 (now 3b). We now also reiterate this point in Section 5 (Discussion):

"As a further caveat of our results, we want to reiterate the fact that all the geographic reconstructions in the simulation study are informed by the true phylogenetic tree of the simulated languages. In reality this phylogeny needs to be inferred from linguistic data, which is a difficult and challenging task. [...]" [lines 514-517]

I would personally expand the explanation for RMSE in the main text (e.g. the Euclidean distance between every location in the posterior distribution and the true root).

We moved the more detailed explanation of the RMSE to the main paper as part of Section S3 (now 3c). [lines 309-311]

Speaking of the posterior distribution, I would be interested to know how many iterations these runs were set to, and how much was discarded as burn-in. These details can of course go in the supplement, alongside the details about the priors (end of section S2).

We added this information (along with the prior settings of the analysis) in the supplement (S3):

"We ran the MCMC for 10^6 iterations and discarded the first 10^5 iterations as burn-in. This might seem low compared to other phylogenetic analyses, but fixing the phylogenetic tree reduces the sampling complexity significantly." [Supplementary Material, lines 92-96]

I would also expand the explanation of the observed trend in the main text; the explanation in the supplement ("the distance between the homeland of an expansion and the mean of the final locations" seems more straightforward to understand than the "mean displacement" in my opinion).

We moved the explanation from the supplement to the main paper (Section 3c). [lines 328-340]

Results:

Page 7 of 14:

o Lines 34-37: I got confused about one aspect: do the authors use a relaxed random walk for the two directional trends? (CDRW and RDRW) - in the supplement both the CDRW and RDRW are mentioned, but in the Results section, only the CDRW is mentioned.

That was indeed an inconsistency that we have missed in the initial manuscript. Thank you very much for pointing this out. We intentionally excluded the RDRW runs from the Results section since they gave the same results as the CDRW runs (which is not surprising, since we only simulate constant directional

trends). But of course, this needs to be explained in the main article and the results should be presented in the supplement.

We now added the results as Figures S6 and S7 to the supplement, along with a brief explanation in Section S5 [Supplementary Material, lines 113-122]. We reference these results in Section 4 (Results) of the main paper. [lines 382-383]

o Line 43: regarding historical data, I think it's important to mention how this is added into the migration scenario in the main text (this would also clarify why the expansion scenario only has a "no historical data" panel in figure 3).

We now explain this in more detail in the supplement (S1):

"In the MigSim scenario we simulated phylogenetic trees according to a birth-death process with birth rate $\lambda = 0.00115$ and death rate $\mu = 0.00023$. This yields trees with an average of $N = 100$ extant languages. The death rate allows us to simulate extinct languages, which we included in some of the experiments. In the Results section of the main article (Section 4) we show three settings:

**No historical data: only the extant languages are used for the reconstruction.*

**Sample age ≤ 500 : include languages that went extinct in the last 500 years.*

**All sample ages: include all languages, extant or extinct.*

We achieved these settings by filtering the extinct languages generated by the birth-death process to only include the specified ages. To ensure that the extinct languages have an observable effect, we defined a minimum fossil number of 10, i.e. we restarted the simulation when the filtering resulted in less than 10 fossils (except for the "no historical data" setting)." [Supplementary Material, lines 39-50]

I enjoyed reading the Discussion, section (a) is very clear and explains important differences between the expansion and migration scenarios, while section (b) puts these theoretical considerations into practice with a welcomed example. My personal preference would be that the three descriptive statistics in section (a) would be explained in more detail, however, in the interest of saving space, I completely understand why their equations/algorithms have been moved to the supplement.

We now briefly introduce each of the statistics in the main article:

"Diversity-space dependence measures the correlation between the spatial expansion and diversification of a clade. A strong correlation implies that clades expanding faster also diversify faster, resulting in a higher number of languages." [lines 436-438]

and

"Finally, tree imbalance quantifies the asymmetry of a tree, i.e. the extent to which lineages in the tree diversify unevenly. " [lines 442-444]

We decided against moving the full explanations to the main article, to save space and because we don't see the statistics as the core message of our study.

Also, I wondered if the authors can comment on the following: the descriptive statistics make sense intuitively, and they hold true when computed over the "true" tree, i.e. a tree resulted from simulating a migration or expansion process; however, in an empirical analysis, the tree would be reconstructed alongside the homeland; so I was wondering: would these statistics be the same when the tree is also an

output of the model? E.g. would the outputted tree be balanced in a migration scenario, or would the model wrongly output an imbalanced tree?

As you correctly point out, in an empirical analysis the phylogenetic tree will be subject to errors/uncertainty, and it is unclear how such errors would affect the descriptive statistics. One source of such errors could be the misspecification in the geographical model, which would give the problematic circular justification (which you are hinting at): Biases in phylogeographic models propagate to errors in the descriptive statistics, which are then used to argue for the validity of the phylogeographic model. This can be easily avoided by not using phylogeographic models in the calculation of the descriptive statistics (and instead infer the trees on linguistic data alone), which we now mention in the paper:

"To avoid an influence of the phylogeographic model on the descriptive statistics, and thereby a potentially circular justification, the phylogenetic tree used in the computation of these statistics should be inferred from linguistic data alone (without a geographic model)." [lines 450-453]

Another source of such errors could be misspecifications in the model of lexical evolution (e.g. borrowing). We now discuss this issue in the article (see below). This is a known challenge in phylogenetic inference and could potentially affect any study relying on the reconstruction of trees from linguistic data. That being said, we would not expect severe effects on the tree shape. Errors in the topology seem to require quite high levels of horizontal transfer [greenhill2009does] and many such errors would need to accumulate in a consistent way to really affect the tree shape.

"As a further caveat of our results, we want to reiterate the fact that all the geographic reconstructions in the simulation study are informed by the true phylogenetic tree of the simulated languages. In reality this phylogeny needs to be inferred from linguistic data, which is a difficult and challenging task. Typically, phylogenetic reconstructions are based on models of language evolution through vertical transfer (inheritance), while horizontal transfer of linguistic features (borrowing) would violate the model assumptions. The severity of this problem, i.e. the effect of borrowing on the reconstruction, depends on the type of data used for the analysis --- for example, core vocabulary has been proposed to be fairly stable to borrowing [greenhill2010shape, bowern2011does]. Thus, wherever possible it is advisable to remove borrowed features from the data. However, simulation studies have shown that Bayesian phylogenetic reconstructions are resilient to moderate levels of borrowing [greenhill2009does]. For the current study, we intentionally avoided these problems by providing the phylogenetic tree in the reconstruction. Our focus is the geographic model underlying Bayesian phylogeography. Using the true phylogeny allows us to isolate biases in the geographic models from potential errors in the phylogeny, propagating to errors in the geographic reconstruction." [lines 514-527]

Reviewer 2

This paper is a valuable and well-executed contribution to the growing literature on the application of Bayesian phylogenetic methods to linguistic data. Work like this probing the suitability and reliability of widely used models for historical linguistic purposes is sorely needed. I recommend the paper be accepted for publication. Below I include some remarks on how I think the paper could be improved, but I do not consider any of them to be addressing serious flaws or shortcomings and do not feel that publication should be contingent upon these changes being made.

Thank you very much for your review. The additional references help to expand the scientific context, the information we added upon your suggestions will help the reader to better understand the study and the further detailed comments made the paper more consistent in general. Also, we appreciate the comments on the publication of our software, which is an easily overlooked issue.

The introduction/overview of phylogeography disregards the existence of discrete phylogeographic methods. This is perhaps understandable/acceptable in the context of the present paper, as only continuous phylogeographic methods have been used thus far in historical linguistic contexts, but it may still be worthwhile clarifying that phylogeography as a field is a little broader than this. Discrete methods are widely used and influential in biogeography, and function especially well in the context of island groups, where species/languages can be modelled as being either present or absent on each individual island. There's an epidemiological example in the Lemey et al paper cited below - not necessarily the best example citation on discrete phylogeography by any means, just one I happened to be immediately aware of.

Lemey P, Rambaut A, Drummond AJ, Suchard MA (2009) Bayesian Phylogeography Finds Its Roots. PLOS Computational Biology 5(9): e1000520. <https://doi.org/10.1371/journal.pcbi.1000520>

We now briefly mention the existence of discrete phylogeographic models, give examples from the literature and argue why these models are less relevant with regards to the context of the present paper:

"Phylogeography can be viewed as a special case of trait evolution, where the changing property is the spatial location, modelled as either a discrete site [35] or a continuous position on a plane [10,34] or sphere [36]. While discrete phylogeographic models are important, e.g. in epidemiology [37,38], they have not been used thus far in historical linguistic contexts, which is why we focus on continuous models" [lines 148-151]

The ExpSim methodology bears some degree of similarity to the the language expansion model described in an earlier paper by Gavin et al (full citation below), and I think it would be appropriate to mention this in the paper and cite that work. I don't doubt for a second that ExpSim was developed independently and don't mean to imply that the authors of that earlier work deserve any credit for the present work - instead I'm just very aware that readers who are interested in using or adapting one model are highly likely to be interested in doing the same to the other, and it's a useful secondary purpose of scientific papers to help "knit the literature together" and enable this kind of idea discovery.

Gavin, M.C., Rangel, T.F., Bowern, C., Colwell, R.K., Kirby, K.R., Botero, C.A., Dunn, M., Dunn, R.R., McCarter, J., Pacheco Coelho, M.T. and Gray, R.D. (2017), Process-based modelling shows how climate and demography shape language diversity. Global Ecol. Biogeogr., 26: 584-591. doi:10.1111/geb.12563

We agree that this kind of references to similar models will be useful to readers. We added the reference in Section 3a (ii):

"This simulation carries some resemblance to the model presented by Gavin et al. [gavin2017process], but with crucial differences in the model: While they simulate language areas sequentially, the ExpSim model simulates splits, which induce a phylogenetic relationship between the languages." [lines 259-262]

Another comment (not added to the paper for reasons of brevity): Another important difference is that Gavin et al. are centrally interested in spatial differences of language density. To achieve that, the size of their area depends on the carrying capacity (which is not a factor in our simulation).

It was not immediately clear to me on my first reading what the "HPD coverage" quantity shown in Figures 2b and 3b actually was, although reading the relevant paragraph on page S4 of the Supporting Information clarified this. I would suggest that perhaps some of this wording be migrated from the SI to the main paper. Even after reading the SI, though, I'm not actually sure exactly how the "HPD region" is defined. I guess it's the convex hull of all the sampled root locations?

We added an explanation to the Supplementary Material:

"That is, we used TreeAnnotator (part of the BEAST toolbox [1]) to compute spatial polygons, which contain 80% (or 95%, respectively) of the probability mass with the highest density, according to a spatial kernel density estimation (KDE) on the posterior samples. The coverage of these HPD polygons is the fraction of simulation runs in which polygons contain the simulated homeland location. ..."

[Supplementary Material, lines 100-111]

I remain a little unclear on a few mechanical details about the ExpSim model, which aren't really essential to appreciating its role in this paper but which nevertheless it would be nice to have explained so readers don't have to resort to reading the code. How are fossil locations generated for languages under this model, when the languages themselves occupy a region of space (like real languages!) while continuous phylogeographic models model language locations as individual points (an unfortunate simplification)?

We added an explanation on how we map simulated areas to point locations in the supplement (S2):

"In order to run the phylogeographic analyses, we eventually represent the simulated language areas by their centre of mass (the tested methods require a point location)." [Supplementary Material, lines 71-73]

We do not simulate fossils in the ExpSim scenario. We added a clarification in Section (4):

"We did not simulate fossils in the ExpSim scenario. Even in the absence of historical information the estimated reconstructions consistently recover the simulated homeland, therefore including fossils was not necessary." [lines 383-386]

When a language grows to its randomly sampled splitting size, how does the split actually happen? Is the new language created as a single cell neighbouring the split language? Or are half of the cells in the splitting language reassigned to the new language?

We now explain the splitting process in the supplement (S2.2):

"When a language splits, the cells of the area are divided along a straight line and each daughter language inherits the cells on one side of this line. To ensure that the daughter languages still comprise reasonably compact areas, we split the previous area along the axis of maximum variance (illustrated in Figure S1)." [Supplementary Material, lines 74-77]

The authors are to be commended on making all the code required for a full reproduction of the present study freely available in a public GitHub repository. I would strongly encourage the authors to add an explicit license to the code so that it is unambiguous that other researchers can adapt it for future use.

Many thanks for pointing this out. We now added a GPLv3 license to the repository.

In principle, this code should also be subject to peer review and I'd have loved to do this myself but time pressures prevented it. I would encourage the authors to consider the prospect of releasing the MigSim and ExpSim tools as separate packages, independent of the exact analyses in this paper, with generalised interfaces and user documentation. Such a tool could be submitted to a software-oriented journal, such as Journal of Open Research Software, to give it a better chance to be scrutinised.

We appreciate this idea and will consider the option to publish the code as a software package in a separate publication.

Minor comments on the writing:

In the first paragraph to begin on page 3, there is a missing author name for reference 18 (it appears currently as "author?" in bold).

We fixed the broken reference. Thank you for pointing this out.

The term "demic process" should be defined immediately after its first use.

We now explain the term immediately after its first use:

"Migration, as we describe it here, is most often a demic process, i.e. the transfer of languages results from a transfer of populations, ..." [lines 61-62]

It's a very small thing to point out, but my earlier academic background has left me a bit of a stickler for correct use of mathematical notation, so I feel compelled (sorry!) to do a little knit-picking regarding section S4.1 of the Support Information. When describing the range of birth- and death-rates used in MigSim, the authors make use of the set membership symbol, but do not use the standard "curly brackets" to denote the set of values. They use "square brackets" - with two values separated by commas between them, these would denote a closed interval, which is indeed a set and hence valid on the right hand side of a set membership symbol, but with three numbers inbetween this just reads as gibberish to a mathematically literate reader.

We appreciate the attention to detail and we now use the standard "curly bracket" notation. [Supplementary Material, lines 145-146]

In a similarly pedantic vein, the authors say they varied the split size between (140,200) and (25,33). This notation denotes open intervals, i.e. the set of all real numbers between, but not including, the endpoints. I suspect the endpoints actually were used in this case (though I could be wrong), however even closed intervals of [140,200] and [25,33] wouldn't be appropriate here because MigSim's split size can only take on integer values. It's probably simpler to just use words here, as indeed is done when describing the default case where split size is sampled between 70 and 100.

We decided on the set notation here as well (i.e. {140, ..., 200}). Using words made the sentence quite convoluted in this case. [Supplementary Material, line 153]

Reviewer 3

This paper aims to study the relevance of phylogeographic methods for reconstructing spatiotemporal patterns of language diversification. The originality of the study is to use simulations of language diversification to test the validity of phylogeographic approaches. The authors simulate expansions and long-distance migrations of populations (languages?) and demonstrate that expansions can be relatively efficiently traced, but migrations cannot be traced at all, especially if no historical data are available. The authors conclude with recommendations on the use of phylogeographic methods to reconstruct the spatial evolution of languages.

The paper is well written, and the statistical analyses seem correct as far as I can judge. I think that the results of this paper may be potentially interesting, but I see several major problems in its present state, that must be addressed prior to publication.

Thank you very much for these very important and insightful comments. We think the points brought up here are crucial to avoid misleading interpretations of our results and confusions about matters of genetic and linguistic evolution.

There is for me a confusion throughout the paper between biological (or genetic) evolution and linguistic evolution. While the two may be correlated in some cases, they can also be completely disconnected when cultural traits are transmitted between populations without involving movements of populations. While this issue is briefly introduced, it should be discussed in much greater depth both in the introduction and in the discussion, especially as the authors mention that this is a crucial point in the discussion of the results (page 4 line 41).

We now elaborate on the issue and why we think the distinction between demic and cultural spread is not central to this article:

"We have seen that both expansion and migration can be driven by demic or cultural processes. However, Bayesian phylogeographic modelling is blind to these processes: reconstruction is based on a model of points moving in space. If the reconstruction is based on genetic data, these points may be readily interpreted as populations. In our case they represent languages, which spread both, through population movements and through language shifts. Which of these was the driver in a specific setting can only be a matter of interpretation in phylogeographic studies. As such, it is not central to this study, where we evaluate the performance of phylogeographic reconstructions itself." [lines 87-94]

Furthermore, we added another caveat in the Discussion, which should address the concerns about assumptions and biases in language phylogenetics:

"As a further caveat of our results, we want to reiterate the fact that all the geographic reconstructions in the simulation study are informed by the true phylogenetic tree of the simulated languages. In reality this phylogeny needs to be inferred from linguistic data, which is certainly not trivial or uncontroversial. Typically, phylogenetic reconstructions are based on models of language evolution through vertical transfer (inheritance), while horizontal transfer of linguistic features (borrowing) would violate the model assumptions. The severity of this problem, i.e. the effect of borrowing on the reconstruction, depends on the type of data used for the analysis – for example, core vocabulary has been proposed to be fairly stable to borrowing [51,52]. Thus, wherever possible it is advisable to remove borrowed features from the data. However, simulation studies have shown that Bayesian phylogenetic reconstructions are

resilient to moderate levels of borrowing [21]. For the current study, we intentionally avoided these problems by providing the phylogenetic tree in the reconstruction. Our focus is the geographic model underlying Bayesian phylogeography. Using the true phylogeny allows us to isolate biases in the geographic models from potential errors in the phylogeny, propagating to errors in the geographic reconstruction." [lines 514-527]

This confusion can also be seen in the terms used by the authors, for example, the term phylogenetic is used many times when there is no analysis or simulation of genetic data in this study. I can quote in the supplementary material the phrase "While diversification is important to simulate phylogenetically related languages". It is not clear to me what "phylogenetically related languages" means.

We added some discussion on the conceptual distinction (see the previous comment). Regarding the terminology of "phylogenetically related languages", we understand that this might be confusing for the reader and we now use "*genealogically related*" instead. In other places, we avoided the term phylogenetic, where we saw fit, but not in cases where "phylogenetic" refers to concrete models or methods (where we see the term as an established loanword from biology). We hope this addresses the concerns of the reviewer.

Furthermore, the title states that it is about "migrations and expansions in human history" whereas it is about migrations and expansion of LANGUAGES. It therefore seems to me essential that the authors clarify throughout the paper whether they are talking about simulations of languages or populations and that they are more consistent in the terms used.

We agree that it would be wrong to equate language history with population history and we can understand that this distinction was not made clear enough in the article. We made revisions throughout the paper, emphasising that reconstructed movements reflect the history of languages (i.e. they could represent either population movement or spread of languages between populations).

We also changed the title to reflect this issue more explicitly. The new title is:

"Can Bayesian phylogeography reconstruct migrations and expansions in linguistic evolution?"

This brings me to my second major point. The originality of this study lies in the simulation of the evolution of languages during an expansion or a migration of populations, but I am unable to assess whether the chosen model of dispersion/diversification is a good representation of this process. I find that the method section of the paper is far too short concerning simulations and that they should be explained in more detail. To my view, part of the supplementary material should be included in the main text.

We agree that the simulations are the core of our study and the explanation of the simulations should not be put off to the Supplementary Material (an unnecessary attempt to keep the paper short). We now moved the Sections S1-S3 from the supplement to the main article (now 3a-3c).

I think it is necessary to better explain the assumptions underlying this model from the point of view of linguistic evolution and if possible, to compare the model developed with other existing models. Being unfamiliar with this literature, I lacked elements to assess the realism of the model proposed to simulate the evolution of language, but it seems odd to me that languages evolve only through dispersal and split.

While language change is clearly a complex process, which would require more sophisticated simulations, this is not the goal of our study (we don't simulate linguistic features at all). We want to evaluate the spatial reconstructions provided by phylogeography. For that we start from the idealised setting where the phylogenetic tree of the languages is correctly inferred, which allows us to focus on assumptions and biases of the geographic model. From the geographic perspective the only relevant information about languages in this set-up is the phylogenetic tree (defined by splits) and the distribution in space (defined by the dispersal). [lines 514-527]

A better justification of the modelling approach is thus necessary. The articles cited in the present manuscript that use phylogeographic approaches to reconstruct language evolution have been criticized precisely on the ground that language evolution cannot be reconstructed in the same way as the evolution of genetic diversity. It seems necessary to me that the authors of the study position themselves in relation to this hypothesis and place their model in a wider context of simulating linguistic evolution.

It is true that the use of phylogenetic models in linguistics has been criticised. But it is also true that "descent with modification along a tree" is the de-facto standard framework for modelling linguistic evolution. The analogies to biological evolution have been discussed in the literature [1][2] and parallels can be found in the principles of evolution, but also in shared issues of phylogenetic models (horizontal transfer, hybridisation, incomplete lineage sorting, etc.; now also addressed in [lines 95-105]). Importantly, this framework has been the basis of many recent studies in historical linguistics. We think a critical reflection of such studies is important on two levels:

- 1) critical reflection of the framework of linguistic evolution.
- 2) critical reflection of the methods within this framework.

This study is clearly rooted in the second category. To evaluate phylogeographic methods, we stay within the framework of tree-like evolution, but we avoid making more specific assumptions about language change, by using fixed phylogenetic trees in the reconstruction.

[1] List, Johann-Mattis, et al. "Unity and disunity in evolutionary sciences: process-based analogies open common research avenues for biology and linguistics." *Biology Direct* 11.1 (2016): 1-17.

[2] Bowern, Claire. "Computational phylogenetics." *Annual Review of Linguistics* 4 (2018): 281-296.

I think that once these major issues have been clarified, it will be easier to evaluate the importance of the findings.

Editor

I agree with the reviewers that your article is an important contribution. However, I also agree that revisions are necessary to indicate the specific contribution within the larger existing field. In particular, I agree that it would help to include more details on the models in the manuscript. I also agree with reviewer 3 that it is necessary to discuss the potential limitations in modelling the transmission of a cultural trait such as language as a genetically inherited trait, rather than taking other transmission modes into account (for example, horizontal transmission is particularly likely to happen among societies in close geographic proximity, which could have a large influence on the inferred patterns –

e.g. see <https://doi.org/10.1098/rstb.2017.0054>). Given that there are no length restrictions at RSOS, I encourage the authors to expand the relevant sections to clarify these relevant points.

1. We tried to make the following points especially clear in the Introduction (and again in the Discussion): The taxonomy of languages does not always align with the genetic relatedness of the speaker populations. This paper is concerned with linguistic phylogeography. Since linguistic phylogeography cannot distinguish between cultural and demic spread of languages, we always must consider both interpretations. We tried to make the terminology consistent throughout the paper.
2. Current phylogenetic models work on the assumption that "descent with modification along a tree" is the default mode of transmission in language evolution (at least for the type of features used in an analysis). We now point to the literature discussing this assumption and we explain why we think that this study is still valid and important [lines 95-105 and 514-527]. We go into more detail about both these concerns in the responses to Reviewer 3.

Further changes

- We have noticed in the revision process that the abstract exceeded the limit of 200 words (as stated in the author guidelines). We shortened it accordingly to adhere to the guidelines.
- Since a very related and relevant study was recently released as a pre-print, we added this reference in Section 2: *"In a recent study, Wichmann et al. [32] compared different phylogeographic software packages and found `no radical differences` in their performance."* [lines 138-139]

Appendix B

Response to the Editor

Comments to the Author:

The authors have done a great job in addressing the reviewers' comments. The revised manuscript now contains the details necessary to understand the approach and explains the assumptions and limitations.

I only have a few suggestions to potentially change some wording in the introduction to help readers who might not be as familiar with phylogeographic approaches:

We are glad that our revisions met the expectations of the reviewers and we thank the editor very much for the additional remarks. They helped to make the introduction more accessible to a broader audience. Below are our responses.

- In Line 101ff you state: "In this article we focus on the geographic component of phylogeographic models and assume a setting where the phylogenetic tree is faithfully reconstructed."

I think it might help if in the introduction you add a brief explanation of how your approach links to empiricists: it seems to be that there exists a dataset of language that diversified from each other for which you have an inferred tree reflecting which languages are related and when they split, and for each language you also know their current location (either as a centre point or as their range) - based on this the question is: "can we infer where speakers of the ancestral form of the language were located". I think making this more explicit will also help readers who are not familiar with simulations to understand your approach of mechanistically reflecting the processes that might generate the data.

We now explain the setting more explicitly. First in Line 19ff, where phylogeography is introduced:

"It [Phylogeography] builds on reconstructed phylogenetic trees, which reflect how languages are related and when they split. Based on the tree and the present-day locations of the languages, a phylogeographic reconstruction attempts to infer the locations of ancestor languages in the past."

And again (with the hint at open databases like D-PLACE) in Line 110ff:

"Empiricists can aim for a similar setting in cases where a phylogenetic tree of the languages under investigation is available (a range of such phylogenies is available on D-PLACE \cite{kirby2016dplace}). Otherwise, the phylogeny needs to be inferred from linguistic data (typically cognate sets). Either way, these reconstructions are not guaranteed to reflect the true genealogy of languages and need to be subject to scrutiny as well."

- In Line 123f you talk about: "accuracy of the phylogeographic reconstruction" and in Line 192 you state: "We evaluate the performance of phylogeographic reconstructions in a simulation study."

The introduction never clarifies what the actual focus is and how you are going to assess accuracy. You could assess whether the location of the common ancestor is placed in roughly the right place, you could assess the false positive and false negative rate of detecting directional trends, you could see whether a migration scenario can be differentiated from an expansion scenario, etc. It would help to specify in the introduction what the key outcome result is that you will focus on and that you will use to assess accuracy. It's present in lines 306ff, but it would help the reader understand how this study relates to their work and the questions they are interested in. This could link again to specify what data exactly you generate in the simulations (see previous comment).

We did mention in Line 35ff (now Line 40ff) that we evaluate the reconstruction of the root location:

"Our evaluation metrics are based on the error between the reconstructed and simulated root location (i.e. homeland). Of course, the remainder of the process, what happened between the root and the tips, is of interest as well, but focusing on the root simplifies the quantification of the reconstruction error and is indicative of the

quality of the whole reconstruction. Furthermore, reconstructing the homeland of a language family takes a very prominent role in many studies in historical linguistics (e.g. [10,11,15])."

We agree that explaining the setting more explicitly will help the reader. Thus we changed the sentence preceding the part above to explain the output of the simulations, the input of the reconstruction and what we evaluate:

"We simulate languages spreading in space and splitting into new languages, based on different historical scenarios. The simulations output recent locations of the languages and a phylogenetic tree, representing their genealogical relationship. The locations and the tree serve as an input to a phylogeographic reconstruction. We test whether this reconstruction can infer the locations of the simulated ancestor languages."

- In Line 106ff you state "We implement two simulation scenarios"

It looks like there are three simulations in total: migration with point locations, expansion with point location, and expansion with geographic areas. Maybe start with "We first implement"

We really only implemented two simulation scenarios, which of course both have variations with different parameter settings. The expansion simulation always simulates areas, but the tested phylogeographic models are point-based, hence we used the centre of mass as a representation of the areas in the reconstruction.

- In Line 92 you state: "languages, which disperse through both population movements and through language shifts. Which of these actually was the driver in a specific setting can only be a matter of interpretation in phylogeographic studies. As such, it is not central to this article"

This seems to assume that the rate of change and distance that can be covered are the same for the two processes. I guess there could be additional noise: for example, movement of populations might be covering larger distances whereas language displacement presumably would involve direct contact. That is, population movements might be more like the migration model you simulate, whereas language shifts might be more like the expansion model?

Indeed, the type of movement (demic or cultural) most likely has an influence on the parameters of the historical processes, for example on the rate of change. In that case we would describe the direction of causality like this:

type of movement (demic/cultural) → parameters of the process (e.g. rate) → reconstruction quality

In the simulation study we directly vary the parameters of the process and test the influence on the reconstruction quality. In that sense we try to be more general and agnostic about the specific causes of different parameter values.

-In the Legend of Figure 5, as pointed out previously by a reviewer, maybe briefly explain what HPD coverage means (in the legend you use the term credible regions rather than HPD coverage).

We adapted the caption of the figure to also refer to "highest probability density (HPD) region" (consistent with the legend in the plot).

- I have not gone through the whole manuscript in detail but you might want to check for small remaining grammatical errors, eg. Line 122 remove the comma in "migration and expansion processes, lead to quite"

We removed the error in Line 122 and proofread the article another time.